# Expectation-Maximization Contrastive Learning for Compact Video-and-Language Representations

**Peng Jin**[1,3*]    **Jinfa Huang**[1,3*]    **Fenglin Liu**[4]    **Xian Wu**[5]    **Shen Ge**[5]    **Guoli Song**[2†]

**David A. Clifton**[4,6]    **Jie Chen**[1,2,3†]

[1]School of Electronic and Computer Engineering, Peking University, China
[2]Peng Cheng Laboratory, Shenzhen, China
[3]AI for Science (AI4S)-Preferred Program, Peking University Shenzhen Graduate School, China
[4]Department of Engineering Science, University of Oxford, UK    [5]Tencent JARVIS Lab, China
[6]Oxford-Suzhou Centre for Advanced Research, Suzhou, China

{jp21, jinfahuang}@stu.pku.edu.cn    {songgl, chenj}@pcl.ac.cn
{shenge, kevinxwu}@tencent.com    {fenglin.liu, david.clifton}@eng.ox.ac.uk

## Abstract

Most video-and-language representation learning approaches employ contrastive learning, e.g., CLIP [53], to project the video and text features into a common latent space according to the semantic similarities of text-video pairs. However, such learned shared latent spaces are not often optimal, and the modality gap between visual and textual representation can not be fully eliminated. In this paper, we propose Expectation-Maximization Contrastive Learning (EMCL) to learn compact video-and-language representations. Specifically, we use the Expectation-Maximization algorithm to find a compact set of bases for the latent space, where the features could be concisely represented as the linear combinations of these bases. Such feature decomposition of video-and-language representations reduces the rank of the latent space, resulting in increased representing power for the semantics. Extensive experiments on three benchmark text-video retrieval datasets prove that our EMCL can learn more discriminative video-and-language representations than previous methods, and significantly outperform previous state-of-the-art methods across all metrics. More encouragingly, the proposed method can be applied to boost the performance of existing approaches either as a jointly training layer or an out-of-the-box inference module with no extra training, making it easy to be incorporated into any existing methods[‡].

## 1   Introduction

Text-video retrieval [76], which aims to fetch relevant videos using textual queries or vice versa, is an important yet challenging cross-modal task. The dominating paradigm of text-video retrieval is contrastive learning [67, 24, 9, 23, 11, 10], which is a commonly adopted framework for video-and-language representation learning. The core idea of contrastive learning is to pull the textual and visual representations of matched text-video pairs together and push the representations of unmatched text-video pairs apart. In this manner, contrastive learning enables neural networks to learn discriminative video-and-language representations.

---

[*]Equal contribution.
[†]Corresponding author: Guoli Song, Jie Chen.
[‡]Code : https://github.com/jpthu17/EMCL

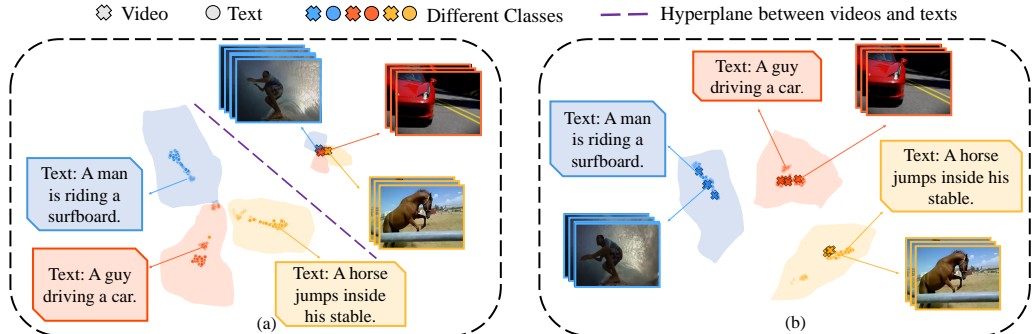

Figure 1: Visualization of video and language representations learned by (a) a recent standard contrastive learning method [21] and (b) our expectation-maximization contrastive learning method. (a) shows that although the representations belonging to the same class are well-clustered, there is still a clear dividing line (see Hyperplane) between video representations and language representations. Besides, the representations with different semantics (i.e., from different semantic classes) overlap with each other. On the contrary, (b) shows that our method can learn more discriminative video-and-language representations. In detail, the video-and-language representations with the same semantics (i.e., belonging to the same semantic class) are well-grouped; and there are clear dividing spaces between different semantic classes.

However, standard contrastive learning has intrinsic limitations for text-video retrieval tasks, since the success of contrastive learning largely depends on the volume and variety of negative samples [9]. Without adequate negative samples, it would be hard to guide the direction of sample learning, and the features could not be contrasted well. As shown in Figure 1a, we randomly select three videos (each video has 20 text captions) from the benchmark dataset [70] and analyze the entire feature space of a recent standard contrastive learning method [21]. Here we visualize the feature space with t-SNE [43]. In an ideal feature space, the instances of the same semantic classes should be close to each other. However, as shown in Figure 1a, we notice that the feature space learned by standard contrastive learning fails to preserve inter-modal semantic relatedness, where videos and texts with the same class semantics are still far away. In other words, the semantic-relevant but modal-different instances can not be grouped together in this feature space. To improve the performance of contrastive learning, recent image-text representation learning methods either increase batch sizes or maintain large data memory banks [9, 11, 44, 67, 23]. These works target to collect sufficient negative samples for contrastive learning. However, such text-video retrieval approaches are relatively expensive due to the incurred extra computational costs, especially on large-scale datasets [57]. So an urgent challenge here is to find an efficient method to learn a semantically relevant feature space.

To efficiently bridge the modality gap and group visual and textual representation according to semantics, we propose to learn a low-rank and compact latent feature space. For this purpose, we propose a novel method named Expectation-Maximization Contrastive Learning (EMCL), which uses the same parametric model to abstract and reconstruct both textual and visual representations. In detail, we find a set of reconstruction bases for subspace representation by estimating the parameters in Expectation-Maximization (EM) algorithm [16]. In the learned subspace, both video and text features are represented with the distributions over the same sets of hidden variables, which can preserve strong semantic relations across modalities. As shown in Figure 1b, our EMCL effectively learns a semantically related subspace which has smaller intra-class variance and larger inter-class variance compared to the previous contrastive learning method (Figure 1a). We further propose EMCL-Net based on the EMCL and apply EMCL-Net to the task of text-video retrieval. In particular, for better adapting the proposed EMCL into the downstream task, our EMCL-Net introduces a parameter initialization strategy of the EM algorithm. Experimental results on three text-video retrieval benchmark datasets, i.e., MSR-VTT [70], ActivityNet [27], and LSMDC [55], show the advantages of the proposed EMCL. The main contributions are as follows:

- We identify the intrinsic limitation of contrastive learning for text-video retrieval, i.e., the cross-modal representation bias could not be fully eliminated via standard contrastive learning approaches.

- To alleviate such limitation, we reformulate the contrastive learning for video-and-language representations into an expectation-maximization iteration manner and propose a plug-and-

play feature projection module named Expectation-Maximization Contrastive Learning (EMCL), which learns the subspace that aims to become semantic-relevant representation.

- Based on our EMCL, we further propose the EMCL-Net, which introduces a parameter initialization strategy for the EM algorithm. Experiments show that our approach achieves state-of-the-art results on three text-video retrieval datasets. More encouragingly, our method can be easily applied to boost the performances of existing approaches either as a jointly training layer or an out-of-the-box inference module with no extra training.

## 2 Related Work

Cross-modal learning is widely studied in many areas, including cross-modal retrieval [76, 67, 24], transfer learning [50, 45], domain adaptation [56, 34], and captioning [39] in which different modalities/domains come from different distributions. The main challenge of cross-modal learning is to use given vision-and-language pairs to learn common representations shared between modalities [38]. Most existing works of text-video retrieval [9, 23, 22] map text and video to the same latent space, where the similarity between them can be directly calculated [20, 6, 63, 17, 65, 66, 7, 15, 73, 52]. In detail, CE [41] introduces a mixture-of-experts method that mixes the features of many different pre-training experts; MMT [21] shows a multi-modal transformer with multiple self-attended layers for video embedding. Recently, contrastive learning methods, e.g., CLIP [53], show great success in advancing the state-of-the-art performances of cross-modal tasks [42]. Contrastive learning methods [19, 5, 23, 44, 10, 40] try to learn data representations from positive and negative pairs, making the representations of positive pairs (usually data augmentations that retain semantic information) have high similarity, and negative pairs (different semantic examples) have low similarity. Inspired by the great success of contrastive learning, we employ CLIP [53] for learning video-and-language representations. In this paper, a positive pair is a matching text-video pair, and a negative pair contains non-matching text and video. However, due to the multi-modal nature and spatial-temporal evolution of video [12, 37], it is essential to capture the most important semantic concept for the video. Meanwhile, the visualization in Figure 1 also shows that the feature space of video-and-language representations has semantically irrelevant redundancy which leads to non-ideal contrast in the feature space. [35] analyzes the non-optimal feature space with modality gap impacts model's performance and fairness. Therefore, we propose to conduct contrastive learning in an expectation-maximization iteration manner to eliminate the non-optimal semantically redundant dimensions, acquiring compact representations.

## 3 Approach

In this section, we first introduce how to reformulate the Contrastive Learning into an expectation-maximization iteration manner, i.e., Expectation-Maximization Contrastive Learning (EMCL). Then, we introduce how to incorporate the proposed EMCL into the neural network for video-and-language representations, which are used to perform the text-video retrieval task.

### 3.1 Preliminaries

**Expectation-Maximization Algorithm** The Expectation-Maximization (EM) algorithm [16] is an iterative optimization strategy, which was originally designed to solve the problem when data are missing in the process of parameter estimation. Briefly, given unobserved hidden variables $\boldsymbol{Z} = \{z_1, z_2, ..., z_N\}$ and observed data sets $\boldsymbol{X} = \{x_1, x_2, ..., x_N\}$ with $N$ samples, the goal of EM algorithm is to estimate the maximum likelihood solution of model parameters $\hat{\theta} = \arg\max \sum_{i=1}^{N} \log \sum_{z_i} p(x_i, z_i; \theta)$. In step E, the EM algorithm calculates the conditional probability expectation $Q_i(z_i) = p(z_i|x_i, \theta)$. In the M step, the likelihood function is maximized to get the revised parameter $\hat{\theta} = \arg\max \sum_{i=1}^{N} \sum_{z_i} Q_i(z_i) \log \frac{p(x_i, z_i; \theta)}{Q_i(z_i)}$.

**Gaussian Mixture Model** The Gaussian Mixture Model (GMM) [54] combines multiple single Gaussian models. Assuming that GMM consists of $K$ Gaussians, given the input $\boldsymbol{X} = \{x_1, x_2, ..., x_N\}$ with hidden variables $\boldsymbol{Y}$, the probability of GMM is as follows:

$$p(\boldsymbol{X}, \boldsymbol{Y}; \mu, \Sigma, \pi) = \prod_{n=1}^{N} \prod_{k=1}^{K} (\pi_k \mathcal{N}(x_n|\mu_k, \Sigma_k))^{y_{n,k}}, \tag{1}$$

**Algorithm 1** The proposed Expectation-Maximization Contrastive Learning, with $T$ iterations of routing. Typically, $T \approx 9$.

---

**Require:** $\boldsymbol{X} \in \mathbb{R}^{2B \times D}$ (Video features and text features), $M \in \mathbb{R}^K$ (Initial Value Maintenance)
1: Initialize $\boldsymbol{\lambda} \in \mathbb{R}^{2B \times K}$ with a splice of $M$ copied $2B$ times
2: **for** routing iteration $t = 1, 2, ..., T$ **do**
3:     $\boldsymbol{Y} \leftarrow \boldsymbol{X}^T \boldsymbol{\lambda} / \sigma$
4:     **for** $j = 1, 2, ..., D$ **do**
5:        $[y_{j,1}, ..., y_{j,K}] \leftarrow \text{Softmax}([y_{j,1}, ..., y_{j,K}])$ # Eq. (6)
6:     **end for**
7:     $\boldsymbol{\lambda} \leftarrow \boldsymbol{X}\boldsymbol{Y}$
8:     **for** subspace $k = 1, 2, ..., K$ **do**
9:        $\lambda_{:,k} \leftarrow \lambda_{:,k} / \sum_{j=1}^{D} y_{j,k}$        # Eq. (7)
10:    **end for**
11: **end for**
12: Reconstruct $\boldsymbol{X} \in \mathbb{R}^{2B \times D}$ with $\boldsymbol{X} = \boldsymbol{\lambda}\boldsymbol{Y}^T$    # Eq. (8)
13: Update $M$ with $m_k = \alpha m_k + (1 - \alpha)\overline{\lambda_{:,k}}$   # Eq. (9)
14: **return** $\boldsymbol{X} \in \mathbb{R}^{2B \times D}, M \in \mathbb{R}^K$

---

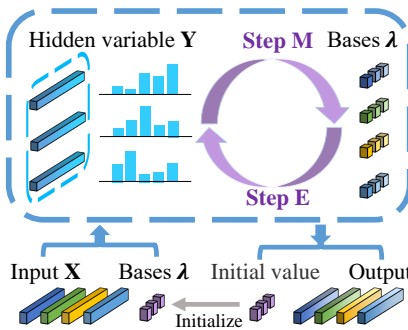

Figure 2: **Module overview.** For clarity, we show the case when the number of subspaces $K = 3$. The input features come from 4 samples, which are colored differently. EMCL projects input features into $K = 3$ subspaces to filter out the redundancy in features.

where $\mathcal{N}(*|\mu_k, \Sigma_k)$ is the probability function of the $k_{\text{th}}$ Gaussian and $\pi_k$ is the prior probability.

Through the E step of the EM algorithm, the estimated value of $y_{n,k}$ can be obtained by:

$$\hat{y}_{n,k} = \frac{\pi_k \mathcal{N}(x_n|\mu_k, \Sigma_k)}{\sum_{k=1}^{K} \pi_k \mathcal{N}(x_n|\mu_k, \Sigma_k)}. \tag{2}$$

In the M step of the EM algorithm, the parameters of the Gaussian mixture model are updated as:

$$\hat{\pi}_k = \frac{\sum_{n=1}^{N} \hat{y}_{n,k}}{N}, \ \hat{\Sigma}_k = \frac{\sum_{n=1}^{N} \hat{y}_{n,k}(x_n - \mu_k)^T(x_n - \mu_k)}{\sum_{n=1}^{N} \hat{y}_{n,k}}, \ \hat{\mu}_k = \frac{\sum_{n=1}^{N} \hat{y}_{n,k}x_n}{\sum_{n=1}^{N} \hat{y}_{n,k}}. \tag{3}$$

### 3.2 Expectation-Maximization Contrastive Learning (EMCL)

In this section, we introduce our EMCL in detail. Representing the features in low-dimensional space is a fundamental method to eliminate the inherent redundancies in features, which is also the goal of our method. We leverage these compact features for contrastive learning to efficiently bridge the gap and improve their performance. The overview of our EMCL is shown in Figure 2 and Algorithm 1.

Mathematically, the subspace can be defined as follows: let $\boldsymbol{X} \in \mathbb{R}^{B \times D}$ be the features, where $B$ is the sample size, $D$ is the dimension of the feature. Suppose that these features are distributed in many unknown linear subspaces, where the intrinsic dimensions of the linear subspaces are less than $D$. We define the union of $K$ semantically related low-dimensional linear subspaces as the semantic subspace. To this end, our method includes two parts: Firstly, we use the EM algorithm [16] to find the bases of the $K$ optimal subspaces, which is called "**Maximum Probability Projection**"; Secondly, we need to re-represent the features in these subspaces, which is called "**Feature Reconstruction**"; Finally, in the "**Contrastive Learning**", the compact features we re-represent are used in contrastive learning to bridge the gap and improve their performance.

**Maximum Probability Projection**   Given original video features $\boldsymbol{C_v} \in \mathbb{R}^{B \times D}$ and original text features $\boldsymbol{C_t} \in \mathbb{R}^{B \times D}$, we combine both video features and text features to be the input data $\boldsymbol{X} \in \mathbb{R}^{2B \times D}$, where $x_{i,j}$ is the $j_{\text{th}}$ coding bit of the $i_{\text{th}}$ sample. In our method, each coding bit $x_{i,j}$ is assigned to a specific subspace, and meanwhile we introduce the hidden variable $\boldsymbol{Y}$, where $y_{j,k}$ represents whether the $k_{\text{th}}$ subspace is selected by $x_{i,j}$. We take the base $\boldsymbol{\lambda}$ of the subspace as the parameter of EM algorithm [16], where $\boldsymbol{\lambda} \in \mathbb{R}^{2B \times K}$ represents the distribution of samples in $K$ semantically related low-dimensional linear subspaces. Given input data $\boldsymbol{X} \in \mathbb{R}^{2B \times D}$ and hidden variable $\boldsymbol{Y} \in \mathbb{R}^{D \times K}$, the complete likelihood of the input feature $\boldsymbol{X}$ can be expressed as:

$$P(\boldsymbol{X}, \boldsymbol{Y}; \boldsymbol{\lambda}) = \prod_{i=1}^{2B} \prod_{j=1}^{D} \prod_{k=1}^{K} (p(x_{i,j}, \lambda_{i,k}))^{y_{j,k}}, \tag{4}$$

where $P(*)$ is the probability function and $p(*)$ is a kernel function. Here we assume that $x_{i,j}$ can be represented by $K$ shared Gaussian distributions like GMM [54]. For simplicity, we freeze $\Sigma$ and $\pi$, and only consider the mean $\lambda = \mu$ of the Gaussian model.

There are many choices for the kernel function $p(\hat{x}, \hat{y})$, such as linear kernel $(\hat{x}^T \hat{y} + c)$, polynomial kernel $(a\hat{x}^T \hat{y} + c)^d$, Gaussian kernel $exp(-\hat{\gamma}\|\hat{x} - \hat{y}\|^2)$ and so on. We find that different kernel functions have slightly different impacts on the final results. For easy implementation, we use Gaussian kernel and rewrite it in a form similar to the attention model [2, 60]. The formula is:

$$p(x_{:,j}, \lambda_{:,k}) = \exp\left(\frac{\sum_{i=1}^{2B} x_{i,j}\lambda_{i,k}}{\sigma}\right), \tag{5}$$

where $\sigma$ is a hyper-parameter to adjust the distribution, similar to the mean and covariance in the Gaussian distribution. In this paper, we define $x_{:,j}$ to be the $j_{\text{th}}$ column of $\boldsymbol{X}$, and $\lambda_{:,k}$ to be the $k_{\text{th}}$ column of $\boldsymbol{\lambda}$.

Reviewing the EM algorithm [16], it estimates the parameters of the model by continuously executing E steps and M steps. In step E, it calculates the conditional probability expectation with current parameters. In step M, it maximizes the likelihood function to update the parameters.

In step E, referring to the Eq. (2), the estimated value of $y_{j,k}$ can be obtained by:

$$y_{j,k} = \frac{p(x_{:,j}, \lambda_{:,k})}{\sum_{k=1}^{K} p(x_{:,j}, \lambda_{:,k})}. \tag{6}$$

In the M step, we update base $\lambda_{i,k}$ according to Eq. (3), which is formulated as:

$$\lambda_{i,k} = \frac{\sum_{j=1}^{D} y_{j,k} x_{i,j}}{\sum_{j=1}^{D} y_{j,k}}. \tag{7}$$

By repeatedly iterating step E and step M, our algorithm forces this set of optimal subspaces to represent the original video features and original text features at the same time. After several iterations, we could keep the information that appears in both the video and the text to remove the redundancy.

**Feature Reconstruction**    When reconstructing features, we use $\boldsymbol{Y}$ and $\boldsymbol{\lambda}$ to linearly reconstruct the original features. Learning from the meaning of $\boldsymbol{Y}$ and $\boldsymbol{\lambda}$ mentioned above, $y_{j,k}$ indicates whether the $k_{\text{th}}$ subspace is selected by $x_{:,j}$ and $\lambda_{:,k}$ represents the base of the $k_{\text{th}}$ subspace. The formula for feature reconstruction is:

$$x_{i,j} = \sum_{k=1}^{K} \lambda_{i,k} y_{j,k}. \tag{8}$$

As shown in Eq. (8), we only use the corresponding base $\lambda_{i,k}$ (for all $k \in [1, K]$) to reconstruct the $i_{\text{th}}$ sample $x_{i,j}$. Therefore, we estimate different $x_{i,j}$ from $K$ subspaces.

**Contrastive Learning**    The above feature reconstruction method generates the low-rank and compact features stripped of redundancy. Therefore, we can leverage these compact features for contrastive learning to efficiently bridge the modality gap and improve task performance.

### 3.3 EMCL for Cross-Modal Learning

In this section, we discuss in detail how to incorporate the EMCL into the neural network for cross-modal downstream tasks. Targeting on the cross-modal tasks, we propose the ECML-Net, which introduces a parameter initialization strategy for the EM algorithm that can establish the connection between batches. At last, we take an important yet challenging cross-modal task, i.e., the text-video retrieval, as an example to illustrate how to employ our EMCL-Net to perform downstream tasks.

**Initial Value Maintenance**    Considering that a corpus of video-based cross-modal tasks usually contains thousands of text-video pairs, it is often not enough to only establish connections between samples within single batches. Therefore, our module builds an inter-batch information transfer mechanism by maintaining a separate initial value $M \in \mathbb{R}^K$. In implementations, each time a new batch of features is fed into the model, the EM algorithm needs to initialize the $\boldsymbol{\lambda}$. Rather than randomly initializing $\boldsymbol{\lambda}$ for each batch, we use $M$ as the initial value of the $\boldsymbol{\lambda}$. For all $i, k$, we

initialize $\boldsymbol{\lambda}$ with $\lambda_{i,k} = m_k$. We update $M$ using an average moving method similar to the Batch Normalization (BN) layer [25]. The moving method is formulated as:

$$m_k = \alpha m_k + (1 - \alpha)\frac{1}{2B}\sum_{i=1}^{2B}\lambda_{i,k}, \tag{9}$$

where $\alpha \in [0, 1]$ is the momentum. Similar to the average moving method in the BN layer, we don't update $M$ in the inference stage. Because the initial value is crucial to the stability of the EM algorithm [1, 33], we need to limit the value range of the initial value. In each iteration, we use the L2 normalization operation to limit the value range of $\boldsymbol{\lambda}$.

**EMCL-Net** For fair comparisons, we follow common practice [42, 13, 62] to extract the video representations of input videos and the language representations of input texts. In detail, for video representations, we first extract the frames from the video clip as the input sequence of video; Then we use ViT [18] to encode the frame sequence, by exploiting the transformer architecture to model the interactions between image patches; Followed by the CLIP [53], the output from the [class] token is used as the frame embedding; Finally, we aggregate the embedding of all frames and obtain the video representation $\boldsymbol{C_v} \in \mathbb{R}^{B \times D}$. For text representation, we directly use the text encoder of CLIP to acquire the text representation $\boldsymbol{C_t} \in \mathbb{R}^{B \times D}$.

As a plug-and-play module, our proposed EMCL is inserted at the end of the video-text encoders. We concatenate video features $\boldsymbol{C_v}$ and text features $\boldsymbol{C_t}$, generating the input data of our method $\boldsymbol{X} = [\boldsymbol{C_v}; \boldsymbol{C_t}] \in \mathbb{R}^{2B \times D}$. Following [14], we add reconstructed features $f_{\text{EMCL}}(\boldsymbol{X})$ by the EMCL and original features $\boldsymbol{X}$ to obtain final text-video representations $\hat{\boldsymbol{X}} = [\hat{\boldsymbol{C_v}}; \hat{\boldsymbol{C_t}}] = \beta f_{\text{EMCL}}(\boldsymbol{X}) + \boldsymbol{X} \in \mathbb{R}^{2B \times D}$ where $\beta$ is the scale factor. Here, $\beta$ is the scale factor which is used to balance the original and new embeddings. By adjusting $\beta$, we can add some flexibility in the reconstructed video-and-language representations for the robustness of the model.

**Training Objective** During the training stage, the EMCL module is trained with the neural network. We use $M$ to initialize the $\lambda$. Then the components $Y$ and $\lambda$ are updated in unsupervised way by iteration, as shown in Algorithm 1. Moreover, we update the initial value $M$ using an average moving method. The goal of the EMCL-Net is to map text and video into a joint representation space to measure the text-video similarity. Following common practice, we use cosine similarity $s(t, v) = \frac{\hat{c}_t^T \hat{c}_v}{\|\hat{c}_t\|\|\hat{c}_v\|}$ as the similarity measure between text $t$ and video $v$, where $\hat{c}_t$ represents the reconstructed text representation of text $t$; and $\hat{c}_v$ represents the reconstructed video representation of video $v$. We train our model with InfoNCE loss [59]:

$$\mathcal{L} = -\frac{1}{2}\left(\frac{1}{B}\sum_{i=1}^{B}\log\frac{\exp(s(t_i, v_i)/\tau)}{\sum_j^B \exp(s(t_i, v_j)/\tau)} + \frac{1}{B}\sum_{i=1}^{B}\log\frac{\exp(s(t_i, v_i)/\tau)}{\sum_j^B \exp(s(t_j, v_i)/\tau)}\right), \tag{10}$$

where $B$ is the batch size and $\tau$ is the temperature hyper-parameter. This loss function maximizes the similarity of positive pairs $s(t_i, v_i)$ and minimizes the similarity of negative pairs.

During the inference stage, given a set of queries (text/video) and a set of candidates (videos/texts), we use the trained $M$ to initialize the $\lambda$. Then $Y$ and $\lambda$ can be updated in an unsupervised way by iteration. The goal of cross-modal retrieval task is to map the query and candidate into a joint video-and-language representation space to measure the query-candidate similarity $s(t, v)$. In this way, we can output the similarity scores for all the input candidates, and take the candidates with the Top-1/5/10/50 similarity scores with input query as the final prediction.

## 4 Experiments

**Datasets, Metrics and Implementation Details** *Datasets.* We conduct the experiments on three popular text-video retrieval datasets, i.e., MSR-VTT [70], ActivityNet Captions [27], LSMDC [55], and follow common practice [42, 13, 62] to pre-process the datasets for fair comparison. In detail, MSR-VTT [70] contains 10,000 videos, each with 20 text descriptions; We follow the 1k-A split [41] with 9,000 videos for training and 1,000 for testing. ActivityNet Captions [27] contains 20,000 videos with multiple sentence descriptions; We report results on the "vall" split (10,009 training, 4,917 testing) as in [21]. LSMDC [55] contains 118,081 video clips from 202 movies; We follow the split of [21] with 1,000 videos for testing.

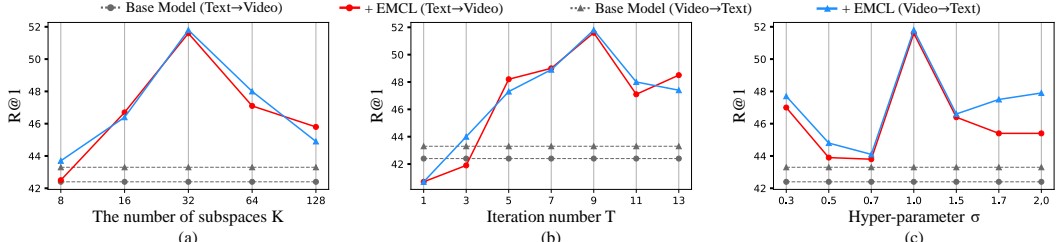

Figure 3: Effect of (a) the number of subspaces $K$; (b) the iteration number $T$; (c) the Hyper-parameter $\sigma$ for our proposed EMCL module with inverted softmax [13, 4].

*Metrics.* We choose standard retrieval metrics: Recall at K (R@K) and Median Rank (MdR) to evaluate the text-to-video and video-to-text retrieval performance.

*Implementation Details.* We utilize the CLIP (ViT-B/32) [53] equipped with Temporal Transformer [42] as pre-trained Bi-Encoder (Base Model). Following previous works [42], the frame length and caption length are 12 and 32 for MSR-VTT and LSMDC. For ActivityNet, a long video retrieval dataset, we set the frame length to 64 and caption length to 64. We follow training schedules from previous works [42, 13, 62]. Concretely, we use the Adam optimizer [26] with a linear warmup. The initial learning rate is 1e-7 for text encoder and video encoder and 1e-4 for other modules. We set the temperature $\tau = 0.01$, $\sigma = 1$, the momentum $\alpha = 0.9$, the number of iterations is set to 9 and the parameter $K$ is set to 32. The network is optimized with the batch size of 128 in 5 epochs. During the training stage, the EMCL module is trained with the neural network. Each time a new batch of features is fed into the model, we use $M$ to initialize the $\lambda$. Then the components $Y$ and $\lambda$ are updated by iteration. Moreover, we update the initial value M using an average moving method. During the inference stage, given a set of queries (text/video) and a set of candidates (videos/texts), we use the trained $M$ to initialize the $\lambda$. Then the components $Y$ and $\lambda$ can be updated by iteration. When the EMCL module is incorporated into trained baselines as an out-of-the-box inference module with no extra training, $\lambda$ is randomly initialized. Then the components $Y$ and $\lambda$ can be updated in an unsupervised way by iteration.

**Comparisons to State-of-the-art** Table 1 shows the results of our method on three text-video retrieval datasets. As we can see, our ECML-Net method consistently outperforms the recently proposed state-of-the-art methods on both text-to-video retrieval and video-to-text retrieval tasks across all datasets and metrics.

**Generalization Analysis** Since the input and output dimensions of the EMCL module are the same, it is model-agnostic and can be applied to features extracted from any language and video encoders. Therefore, we further equip our EMCL with three strong baseline models, i.e., MMT [21], CLIP4Clip [42], DCR [62], and evaluate the performance of these models on the MSR-VTT datasets. Specifically, we insert the EMCL module at the end of the video-text encoders. Table 2 shows that our EMCL can be applied to successfully boost all baselines either as a jointly training layer or an out-of-the-box inference module with no extra training. Overall, our approach can boost the baselines with the most significant improvement up to 3.5% and 4.2% for text-to-video task and video-to-text task in terms of R@1 metric, respectively. The significant improvements demonstrate the generalization ability of EMCL.

**Comparisons to Other Baseline Methods** We further compare our method with other baseline methods in Table 3. PCA [58] is a popular method for finding salient features shared by the two modalities so that the modality gap can be reduced to a certain extent. "Transformer" represents using a shared transformer between two modalities, which can explicitly project the data from two modalities into a shared space. "Fully Connected Layers" represents using two shared fully connected layers between two modalities. "Sparse Autoencoders" is a common sparse autoencoder [46], which reduces the average response of the encoding layer and learns the compact representation. Different from these representative dimensionality reduction methods, the motivation of our method is to use contrastive learning to preserve video-text semantic relatedness. By maximizing the joint posterior probability of video and text, we find a semantically related subspace for compact video-text representation. In contrast, PCA and its variants are feature dimension reduction methods, which aim to maximize the variance of projected data and cannot guarantee semantic relationships. Sparse autoencoder reduces the average response of the encoding layer for sparsity, which may potentially

Table 1: Comparisons to current state-of-the-art methods on the MSR-VTT [70], ActivityNet [27] and LSMDC [55] datasets. "↑" denotes higher is better. "↓" denotes lower is better. †† denotes employing inverted softmax [13, 4].

(a) Retrieval performance on the **Text->Video** task.

| Methods | Pre-trained weights | MSR-VTT [70] | | | | ActivityNet [27] | | | | LSMDC [55] | | | |
|---|---|---|---|---|---|---|---|---|---|---|---|---|---|
| | | R@1↑ | R@5↑ | R@10↑ | MdR↓ | R@1↑ | R@5↑ | R@50↑ | MdR↓ | R@1↑ | R@5↑ | R@10↑ | MdR↓ |
| JSFusion [75] | - | 10.2 | 31.2 | 43.2 | 13.0 | - | - | - | - | 9.1 | 21.2 | 34.1 | 36.0 |
| CE [41] | GPT-1 | 20.9 | 48.8 | 62.4 | 6.0 | 18.2 | 47.7 | 91.4 | 6.0 | 11.2 | 26.9 | 34.8 | 25.3 |
| MMT [21] | BERT-Base | 24.6 | 54.0 | 67.1 | 4.0 | 22.7 | 54.2 | 93.2 | 5.0 | 13.2 | 29.2 | 38.8 | 21.0 |
| Support-Set [49] | T5-Base | 27.4 | 56.3 | 67.7 | 3.0 | 26.8 | 58.1 | 93.5 | 3.0 | - | - | - | - |
| T2VLAD [64] | BERT-Base | 29.5 | 59.0 | 70.1 | 4.0 | 23.7 | 55.5 | 93.5 | 4.0 | 14.3 | 32.4 | 42.2 | 16.0 |
| CLIP4Clip [42] | CLIP (ViT-B/32) | 44.5 | 71.4 | 81.6 | 2.0 | 40.5 | 72.4 | **98.1** | 2.0 | 22.6 | 41.0 | 49.1 | 11.0 |
| EMCL-Net (Ours) | CLIP (ViT-B/32) | 46.8 | 73.1 | 83.1 | 2.0 | 41.2 | 72.7 | **98.1** | 2.0 | 23.9 | 42.4 | 50.9 | 10.0 |
| EMCL-Net (Ours)†† | CLIP (ViT-B/32) | **51.6** | **78.1** | **85.3** | **1.0** | **50.6** | **78.7** | **98.1** | **1.0** | **25.9** | **46.4** | **53.7** | **8.0** |

(b) Retrieval performance on the **Video->Text** task.

| Methods | Pre-trained weights | MSR-VTT [70] | | | | ActivityNet [27] | | | | LSMDC [55] | | | |
|---|---|---|---|---|---|---|---|---|---|---|---|---|---|
| | | R@1↑ | R@5↑ | R@10↑ | MdR↓ | R@1↑ | R@5↑ | R@50↑ | MdR↓ | R@1↑ | R@5↑ | R@10↑ | MdR↓ |
| CE [41] | GPT-1 | 20.6 | 50.3 | 64.0 | 5.3 | 17.7 | 46.6 | 90.9 | 6.0 | - | - | - | - |
| MMT [21] | BERT-Base | 24.4 | 56.0 | 67.8 | 4.0 | 22.9 | 54.8 | 93.1 | 4.3 | 12.1 | 29.3 | 37.9 | 22.5 |
| Support-Set[49] | T5-Base | 26.6 | 55.1 | 67.5 | 3.0 | 25.5 | 57.3 | 93.5 | 3.0 | - | - | - | - |
| T2VLAD [64] | BERT-Base | 31.8 | 60.0 | 71.1 | 3.0 | 24.1 | 56.6 | 94.1 | 4.0 | 14.2 | 33.5 | 41.7 | 17.0 |
| CLIP4Clip [42] | CLIP (ViT-B/32) | 42.7 | 70.9 | 80.6 | 2.0 | 42.5 | 74.1 | 98.1 | 2.0 | 20.8 | 39.0 | 48.6 | 12.0 |
| EMCL-Net (Ours) | CLIP (ViT-B/32) | 46.5 | 73.5 | 83.5 | 2.0 | 42.7 | 74.0 | 98.3 | 2.0 | 22.2 | 40.6 | 49.2 | 12.0 |
| EMCL-Net (Ours)†† | CLIP (ViT-B/32) | **51.8** | **80.2** | **88.0** | **1.0** | **50.6** | **78.9** | **98.4** | **1.0** | **26.7** | **44.7** | **54.4** | **8.0** |

Table 2: Generalization analysis of our EMCL on the MSR-VTT dataset [70]. We equip our EMCL with three strong contrastive learning baselines. † denotes our own re-implementation of baselines; ‡ denotes the EMCL is trained jointly with the baselines from scratch; § denotes the EMCL is incorporated into trained baselines as an out-of-the-box inference module with no extra training. We conducted 5 runs with different seeds for all experiments, the t-tests indicate that $p < 0.01$. The (+Number) denotes the absolute improvements.

| Methods | Text->Video | | | Video->Text | | |
|---|---|---|---|---|---|---|
| | R@1↑ | R@5↑ | R@10↑ | R@1↑ | R@5↑ | R@10↑ |
| MMT [21]† | 25.9 | 54.8 | 68.5 | 26.0 | 58.2 | 69.3 |
| + EMCL (Ours)§ | 26.2 (+0.3) | 57.2 (+2.4) | **70.8 (+2.3)** | 27.2 (+1.2) | **59.8 (+1.6)** | **70.4 (+1.1)** |
| + EMCL (Ours)‡ | **27.1 (+1.2)** | **57.6 (+2.8)** | 70.5 (+2.0) | **27.8 (+1.8)** | 59.3 (+1.1) | 69.8 (+0.5) |
| CLIP4Clip [42]† | 43.4 | 70.4 | 78.5 | 42.4 | 68.6 | 79.2 |
| + EMCL (Ours)§ | 44.6 (+1.2) | 71.4 (+1.0) | 79.5 (+1.0) | 45.0 (+2.6) | 71.2 (+2.6) | 79.2 (+0.0) |
| + EMCL (Ours)‡ | **46.9 (+3.5)** | **72.5 (+2.1)** | **81.8 (+3.3)** | **46.6 (+4.2)** | **73.3 (+4.7)** | **82.3 (+3.1)** |
| DCR [62]† | 46.8 | 72.6 | 82.6 | 45.8 | 72.1 | 82.1 |
| + EMCL (Ours)§ | 47.5 (+0.7) | **74.9 (+2.3)** | 83.8 (+1.2) | 45.9 (+0.1) | **73.6 (+1.5)** | **83.5 (+1.4)** |
| + EMCL (Ours)‡ | **48.0 (+1.2)** | 73.7 (+1.1) | 83.0 (+0.4) | 45.9 (+0.1) | 73.5 (+1.4) | **83.5 (+1.4)** |

discard semantic information. In addition, our method has linear complexity $\mathcal{O}(BDK)$ and does not require additional training. In contrast, the complexity of PCA is $\mathcal{O}(D^3)$ ($D > B$ and $D > K$), and the autoencoder requires additional training.

**Ablative Analysis** *Effect of the parameter initialization strategy.* As shown in Table 4, with or without parameter initialization strategy, our method can successfully promote the base model. It is worth noting that the EM algorithm [16] is sensitive to initial values [1, 33]. In other words, the convergence of EM algorithm depends mainly on initial parameters. Random initialization leads to large fluctuations in convergence results. This fatal flaw limits the performance of our module. Fortunately, with the proposed parameter initialization strategy, our EMCL method surpasses the base model by a large margin with 3.5% R@1 and 1.4% R@1 on the text-to-video and video-to-text tasks, respectively, proving the effectiveness of our parameter initialization strategy used for EMCL-Net.

*Effect of the number of subspaces.* In Figure 3a, we show the effect of the number of subspaces $K$. On the one hand, we find that fewer subspaces mean fewer semantic centers, which limits our module's ability to reconstruct the features. On the other hand, a larger number of subspaces requires more training data, which increases the cost of our model learning. We set the center size $K = 32$ to achieve the best performance in practice.

Table 3: Comparisons to other baseline methods on MSR-VTT dataset [70]. We perform the analysis on the **Text->Video** task. $B$ is the sample size. $D$ is the dimension of the original feature. $K$ is the number of subspaces. $^{\dagger\dagger}$ denotes employing inverted softmax [13, 4].

| Methods | Complexity | | R@1↑ |
| --- | --- | --- | --- |
| | Time | Space | |
| Base Model | - | - | 42.4 |
| + PCA | $\mathcal{O}(D^3)$ | $\mathcal{O}(D^2)$ | 36.0 |
| + Transformer | $\mathcal{O}(B^2D)$ | $\mathcal{O}(D^2)$ | 41.3 |
| + Fully Connected Layers | $\mathcal{O}(BDK)$ | $\mathcal{O}(DK)$ | 42.1 |
| + Sparse Autoencoders | $\mathcal{O}(BDK)$ | $\mathcal{O}(DK)$ | 43.8 |
| + EMCL (Ours) | $\mathcal{O}(BDK)$ | $\mathcal{O}(DK)$ | 46.8 |
| + EMCL (Ours)$^{\dagger\dagger}$ | $\mathcal{O}(BDK)$ | $\mathcal{O}(DK)$ | **51.6** |

Table 4: Effect of the Parameter Initialization Strategy in our EMCL. $^{\dagger\dagger}$ denotes employing inverted softmax [13, 4].

| Methods | Initialization | Text->Video | | | |
| --- | --- | --- | --- | --- | --- |
| | | R@1↑ | R@5↑ | R@10↑ | MdR↓ |
| Base Model | - | 42.4 | 70.8 | 80.6 | 2.0 |
| + EMCL | - | 43.3 | 71.8 | 81.6 | 2.0 |
| + EMCL | ✓ | 46.8 | 73.1 | 83.1 | 2.0 |
| + EMCL$^{\dagger\dagger}$ | ✓ | **51.6** | **78.1** | **85.3** | **1.0** |

| Methods | Initialization | Video->Text | | | |
| --- | --- | --- | --- | --- | --- |
| | | R@1↑ | R@5↑ | R@10↑ | MdR↓ |
| Base Model | - | 43.2 | 70.0 | 81.1 | 2.0 |
| + EMCL | - | 45.1 | 72.8 | 82.3 | 2.0 |
| + EMCL | ✓ | 46.5 | 73.5 | 83.5 | 2.0 |
| + EMCL$^{\dagger\dagger}$ | ✓ | **51.8** | **80.2** | **88.0** | **1.0** |

Table 5: Comparisons with state-of-the-art methods for video captioning on MSR-VTT dataset [70]. We report BLEU-4 [48], METEOR [3], ROUGE-L [36], and CIDEr [61] metrics. All methods only use image modality as input.

| Methods | BLEU-4↑ | METEOR↑ | ROUGE-L↑ | CIDEr↑ |
| --- | --- | --- | --- | --- |
| STG-KD [47] | 40.5 | 28.3 | 60.9 | 47.1 |
| ORG-TRL [79] | 43.6 | 28.8 | 62.1 | 50.9 |
| MGCMP [8] | 41.7 | 28.9 | 62.1 | 51.4 |
| OpenBook [78] | 42.8 | 29.3 | 61.7 | 52.9 |
| ARB-ACL [32] | 42.6 | 28.9 | 61.5 | 51.3 |
| DCD [72] | 43.4 | 29.6 | 61.8 | 52.8 |
| Base model$_{cap}$ | 45.2 | 29.8 | 63.0 | **54.6** |
| + EMCL (Ours) | **45.3** | **30.2** | **63.2** | **54.6** |

Table 6: Comparisons with state-of-the-art methods for video question answering on MSRVTT-QA dataset [69].

| Methods | Accuracy↑ |
| --- | --- |
| ClipBERT [28] | 37.4 |
| VGT [68] | 39.7 |
| VQA-T [71] | 41.5 |
| SiaSamRea [74] | 41.6 |
| MERLOT [77] | 43.1 |
| Co-Tokenization [51] | 45.7 |
| Base model$_{qa}$ | 45.0 |
| + EMCL (Ours) | **45.8** |

*Effect of the iteration number.* In Figure 3b, we show the influence of EM iteration number $T$. Overall performance improves slightly before leveling off. We find that the algorithm has converged when the number of iterations is 9, so we set the number of iterations to 9 as default.

*Hyper-parameter selection.* The parameter $\sigma$ is hyper-parameter to adjust the distribution (Eq. 5), similar to the mean and covariance in the Gaussian distribution. We evaluate the scale range setting $\sigma \in [0.3, 2.0]$ as shown in Figure 3c. We find that R@1 is improved from 43.8% to 51.6% when $\sigma = 0.7$ and saturated with $\sigma = 1$. As a result, we adopt $\sigma = 1$ to achieve the best performance.

**Generalize to other tasks** *Video captioning.* The purpose of video captioning is to describe the content of the video in fluent sentences. "Base model$_{cap}$" uses CLIP [53] to extract video features and is trained with cross-entropy loss. To generate higher-quality sentences, we apply EMCL between video features and ground-truth text features. As shown in Table 5, using EMCL brings significant improvements on caption quality, e.g., gaining a relative improvement of 0.4% at METEOR.

*Video question answering.* Visual question answering requires the model to predict an answer using visual information [30, 29]. We use the target vocabulary for MSRVTT-QA dataset [69], and train a fully connected layer on top of the final language features to classify the answer. "Base model$_{qa}$" uses CLIP [53], a transformer-based [18, 31] visual-language pre-training model, to extract video-and-language features and is trained with cross-entropy loss. To learn compact video-and-language representations, we apply EMCL between video features and question features. Table 6 shows that EMCL can be applied to boost video question answering successfully and boost the baseline with an improvement up to 0.8%.

**Qualitative Analysis** *Analysis of EMCL iterative process.* As shown in Figure 4a, reducing the intra-class variance makes videos and texts belonging to the same semantic class gather, and increasing the inter-class variance makes those belonging to different semantic classes separate from each other. The two conclusions we get from Figure 4b and Figure 4c are as follows. (1) EMCL module reduces the intra-class variance of the same semantic classes and increases the inter-class variance of different semantic classes. (2) The number $K$ of subspaces has a great influence on the EMCL module. The effect of the EMCL module is limited if $K$ is too small, while the intra-class variance increases if $K$ is too large.

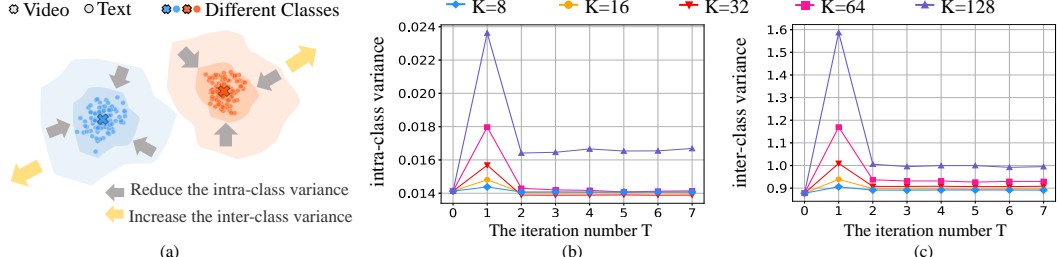

Figure 4: Visualization of EMCL iterative process. (a) shows the influence of reducing the intra-class variance and increasing the inter-class variance on the feature space. (b) shows the relationship between the intra-class variance and the iteration number on the MSR-VTT dataset. (c) shows the relationship between the inter-class variance and the iteration number on the MSR-VTT dataset.

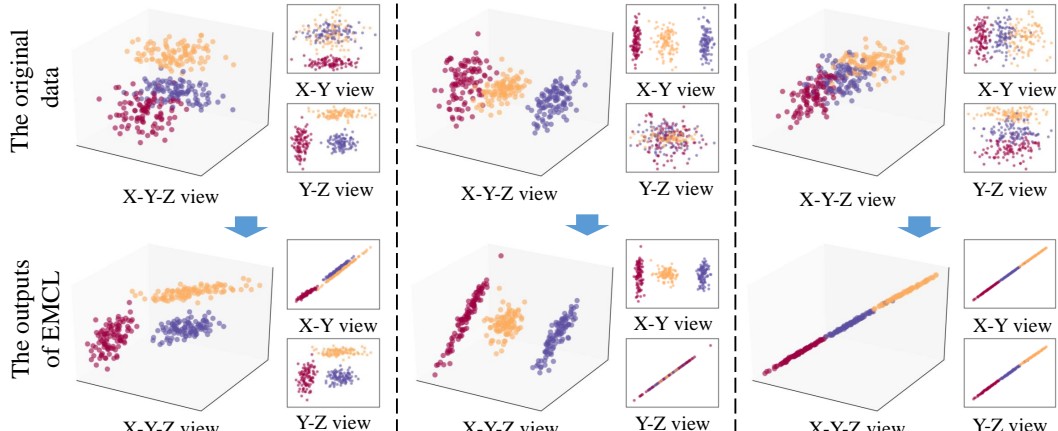

Figure 5: Visualization of our EMCL. We randomly generate a set of 3D data containing 3 types of samples corresponding to 3 different text-video semantic centers. The figure on the top is the visualization of the original data, and the figure on the bottom is the visualization of EMCL's outputs.

*Visualization.* To better understand EMCL, we provide the visualization form of both the original representations and the filtered representations. We notice that the EMCL module eliminates the redundant dimensions, and the features reconstructed by our module are very compact in the feature space. As shown in Figure 5, even though features have intense noise in the redundancy dimension, the EMCL module still learns semantic information. The EMCL module forces the differences between classes in subspace to be more obvious than original, which is helpful for the contrastive learning to learn the semantic centers shared by videos and texts.

## 5 Conclusion

In this paper, we studied the intrinsic limitation of classic contrastive learning for text-video retrieval. We found that the contrastive method in the entire representation space fails to preserve inter-modal semantic relatedness, which makes the features gather or separate in the subspace which is irrelevant to semantics. To mitigate this effect, we propose to directly learn the subspace that is related to shared semantics and do contrastive learning in it. By learning the subspaces related to semantics, we are able to learn the common semantic center of video and text in the semantic subspace. Further, it is worth noting that our method could be applied for other contrastive learning tasks, which include similar samples containing redundant dimensions or with a limited number of negative samples.

**Acknowledgements** This work is supported by the Nature Science Foundation of China (No. 61972217, 62081360152, 62006133, 32071459), Guangdong Basic and Applied Basic Research Foundation (No.2019B1515120049) and Guangdong Science and Technology Department (No. 2020B1111340056). Also, this work is supported in part by the National Institute for Health Research (NIHR) Oxford Biomedical Research Centre; an InnoHK Project at the Hong Kong Centre for Cerebro-cardiovascular Health Engineering; and the Pandemic Sciences Institute, University of Oxford, Oxford, UK.

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
