# Supplementary Material for "Expectation-Maximization Contrastive Learning for Compact Video-and-Language Representations"

**Peng Jin**[1,3][*]    **Jinfa Huang**[1,3][*]    **Fenglin Liu**[4]    **Xian Wu**[5]    **Shen Ge**[5]    **Guoli Song**[2][†]

**David A. Clifton**[4,6]    **Jie Chen**[1,2,3][†]

[1]School of Electronic and Computer Engineering, Peking University, China
[2]Peng Cheng Laboratory, Shenzhen, China
[3]AI for Science (AI4S)-Preferred Program, Peking University Shenzhen Graduate School, China
[4]Department of Engineering Science, University of Oxford, UK    [5]Tencent JARVIS Lab, China
[6]Oxford-Suzhou Centre for Advanced Research, Suzhou, China

{jp21, jinfahuang}@stu.pku.edu.cn    {songgl, chenj}@pcl.ac.cn
{shenge, kevinxwu}@tencent.com    {fenglin.liu, david.clifton}@eng.ox.ac.uk

## A    Additional Discussions

**Potential negative societal impacts**    Although our work improves the performance of text-video retrieval, but may reduce the difficulty of cross-modal retrieval of sensitive information on the network. It may raise challenges to protecting information security.

**Limitations of our work**    Iterative approaches are sensitive to initialization and parameters such as the dimensions and the number of subspaces. In our work, although we use the L2 normalization operation to limit the value range of the parameters, the EM algorithm [3] may still converge to bad results. At the same time, the selection of the number of subspaces also has a relatively significant impact on the model effect. In the future, it is still a promising research direction to explore more stable subspace search algorithms.

## B    Datasets and Implementation Details

**Datasets**    **MSR-VTT** [18] contains 10,000 YouTube videos, each with 20 text descriptions. We follow the training protocol in [10, 5, 12] and evaluate on text-to-video and video-to-text search tasks on the 1K-A testing split with 1,000 video or text candidates defined by [20]. **ActivityNet Captions** [6] dataset consists densely annotated temporal segments of 20,000 YouTube videos. Following [5, 13, 16], we concatenate descriptions of segments in a video to construct "video-paragraph" for retrieval. We use the 10K training split to train or finetune the model and report the performance on the 5K "val1" split. **LSMDC** [15] contains 118,081 video clips from 202 movies. We follow the split of [5] with 1,000 videos for testing. **MSRVTT-QA** [17] is based on the MSR-VTT dataset [18], and has 243,000 VideoQA pairs.

**Implementation Details**    The EMCL module is trained with the neural network. During the training stage, the EMCL module is trained with the neural network. Each time a new batch of features is fed into the model, we use $M$ to initialize the $\lambda$. Then the components $Y$ and $\lambda$ are updated by iteration. Moreover, we update the initial value M using an average moving method. During the inference stage,

---

[*]Equal contribution.
[†]Corresponding author: Guoli Song, Jie Chen.

36th Conference on Neural Information Processing Systems (NeurIPS 2022).

Table 1: Generalization analysis of our EMCL on the ActivityNet [6] and LSMDC [15]. We equip our EMCL with two strong contrastive learning baselines. † denotes our own re-implementation of baselines; ‡ denotes the EMCL is trained jointly with the baselines from scratch; § denotes the EMCL is incorporated into trained baselines as an out-of-the-box inference module with no extra training. We conducted 5 runs with different seeds for all experiments, and the t-tests indicate that $p < 0.01$. The (+Number) denotes the absolute improvements. "↑" denotes higher is better. "↓" denotes lower is better.

(a) Retrieval performance on the ActivityNet [6] dataset.

| Methods | Text->Video | | | Video->Text | | |
|---|---|---|---|---|---|---|
| | R@1↑ | R@5↑ | R@10↑ | R@1↑ | R@5↑ | R@10↑ |
| MMT [5]† | 22.3 | 55.8 | 71.0 | 23.2 | 56.6 | 71.6 |
| + EMCL (Ours)§ | 23.3 (+1.0) | 56.4 (+0.6) | 71.2 (+0.2) | 24.6 (+1.4) | 56.8 (+0.2) | 72.5 (+0.9) |
| + EMCL (Ours)‡ | **25.5 (+3.2)** | **57.3 (+1.5)** | **72.1 (+1.1)** | **25.6 (+2.4)** | **57.7 (+1.1)** | **72.9 (+1.3)** |

(b) Retrieval performance on the LSMDC [15] dataset.

| Methods | Text->Video | | | Video->Text | | |
|---|---|---|---|---|---|---|
| | R@1↑ | R@5↑ | R@10↑ | R@1↑ | R@5↑ | R@10↑ |
| MMT [5]† | 13.1 | 29.6 | 40.4 | 12.1 | 29.2 | 40.1 |
| + EMCL (Ours)§ | 13.9 (+0.8) | 30.3 (+0.7) | **42.4 (+2.0)** | 12.5 (+0.4) | 29.9 (+0.7) | **40.8 (+0.7)** |
| + EMCL (Ours)‡ | **14.6 (+1.5)** | **32.5 (+2.9)** | 42.1 (+1.7) | **14.0 (+1.9)** | **31.5 (+2.3)** | 40.7 (+0.6) |

Table 2: Effect of the scale factor $\beta$ on MSR-VTT [18] dataset with inverted softmax [2, 1].

| | Text->Video | | | | Video->Text | | | |
|---|---|---|---|---|---|---|---|---|
| | R@1↑ | R@5↑ | R@10↑ | MdR↓ | R@1↑ | R@5↑ | R@10↑ | MdR↓ |
| $\beta = 0.0$ | 42.4 | 70.8 | 80.6 | 2.0 | 43.2 | 70.0 | 81.1 | 2.0 |
| $\beta = 1.0$ | 43.6 | 72.1 | 81.4 | 2.0 | 43.2 | 70.8 | 80.8 | 2.0 |
| $\beta = 2.0$ | 45.1 | 69.9 | 79.7 | 2.0 | 45.9 | 71.0 | 78.9 | 2.0 |
| $\beta = 3.0$ | **51.6** | **78.1** | **85.3** | **1.0** | **51.8** | **80.2** | **88.0** | **1.0** |
| $\beta = 4.0$ | 41.5 | 71.3 | 81.4 | 2.0 | 44.2 | 72.1 | 82.8 | 2.0 |

given a set of queries (text/video) and a set of candidates (videos/texts), we use the trained $M$ to initialize the $\lambda$. Then the components $Y$ and $\lambda$ can be updated by iteration. When the EMCL module is incorporated into trained baselines as an out-of-the-box inference module with no extra training, $\lambda$ is randomly initialized. Then the components $Y$ and $\lambda$ can be updated in an unsupervised way by iteration. We utilize the CLIP (ViT-B/32) [14] as pre-trained Bi-Encoder. The temporal transformer is composed of 4-layer blocks, each including 8 heads and 512 hidden channels, and is initialized from the CLIP's text encoder. Following CLIP4Clip [11], the frame length and caption length are 12 and 32 for MSR-VTT and LSMDC. For ActivityNet, a long video retrieval dataset, we set the frame length and caption length to 64 and 64. We follow training schedules from CLIP4Clip [11]. Concretely, we use the Adam optimizer with a linear warmup. The initial learning rate is 1e-7 for text encoder and video encoder and 1e-4 for other modules. We set the temperature $\tau = 0.01$, $\sigma = 1$, the momentum $\alpha = 0.9$, the number of iterations is set to 9 and the parameter $K$ is set to 32. The network is optimized with the batch size of 128 in 5 epochs. All experiments are performed on V100 GPUs.

**The experiment setup in "Comparisons to other baseline methods"**   We chose four baselines, e.g., PCA, "Transformer", "Fully Connected Layers" and "Sparse Autoencoders". For all methods, we concatenate video features $C_v$ and text features $C_t$, generating the input data $X = [C_v; C_t] \in \mathbb{R}^{2B \times D}$. Finally, we add reconstructed features $f(X)$ and original features $X$ to obtain final text-video representations $\hat{X} = [\hat{C}_v; \hat{C}_t] = f(X) + X \in \mathbb{R}^{2B \times D}$. All networks are optimized with the batch size of 128 in 5 epochs. In "PCA", we adopt PCA at the end of the video-text encoders. We use PCA to reduce the dimensions of the original features from 512 to 32, then restore them to 512 dimensions. In "Transformer", we pass the original features through a common Transformer where the inner-layer has a dimensionality of 512. In "Fully Connected Layers", we pass the original features through a common feed-forward network where the inner-layer has a dimensionality of 256, and the activation function is Relu. In "Sparse Autoencoders", we adopt Sparse Autoencoders at the end of the video-text encoders. The inner-layer has a dimensionality of 256, and the activation function is Sigmoid. We reduce the average response of the encoding layer to $\rho = 0.05$.

Table 3: We adopt the EMCL to other contrastive learning tasks, such as Self-supervised Visual reprentation (Self-supervised), Few-shot Image Classification (5way1shot) and Zero-shot Long Video Classification (Zero-shot LVC).

| Task | Method | Dataset | Acc | +EMCL |
|---|---|---|---|---|
| Self-supervised | SimCLR | CIFAR10 | 93.36 | **93.52** |
| 5way1shot | MAML | miniImageNet | 47.5 | **47.7** |
| Zero-shot LVC | Clip4clip | ActivityNet | 51.7 | **52.3** |

Figure 1: Visualization of semantic similarity in semantic subspace based on MMT+EMCL. We take Video 7102 in the MSR-VTT dataset as an example. At the top and bottom are video frames in Video 7102, with text describing the video in the middle. We connect the video frame and the word that are closest in the semantic subspace with a line and place their similarity estimation besides the line.

**The experiment setup in "Generalize to other tasks"** *Video captioning.* The purpose of video captioning is to describe the content of the video in fluent sentences. "Base model$_{cap}$" uses CLIP [14] to extract video features and is trained with cross-entropy loss in 50 epochs. Our framework is based on DCD [19]. We refer the reader to DCD [19] for more detail.

*Video question answering.* Visual question answering requires the model to predict an answer using visual information [8, 7]. We use the target vocabulary for MSRVTT-QA dataset [17], and train a fully connected layer on top of the final language features to classify the answer. "Base model$_{qa}$" uses CLIP [14], a transformer-based [4, 9] visual-language pre-training model, to extract video-and-language features and is trained with cross-entropy loss. We use the Adam optimizer with a linear warmup. The initial learning rate is 1e-7 for text encoder and video encoder and 1e-4 for other modules. The network is optimized with the batch size of 32 in 5 epochs.

## C  Additional Experiments

**Additional Generalization Analysis** To further verify the generalization of our method, we test it on other datasets such as ActivityNet [6] and LSMDC [15]. As a plug-and-play module, our approach EMCL can be easily integrated into existing contrastive learning methods. Therefore, we further equip our EMCL with the baseline model, i.e., MMT [5], and evaluate the performance of the model on the ActivityNet and LSMDC datasets. Table 1 shows that our EMCL can be applied to successfully boost the baseline either as a jointly training layer or an out-of-the-box inference module with no extra training. For ActivityNet, our approach can boost the baseline with the improvement up to 3.2% and 2.4% for text-to-video task and video-to-text task in terms of R@1 metric, respectively. For LSMDC, our approach can boost the baseline with the improvement up to 1.5% and 1.9% for text-to-video task and video-to-text task in terms of R@1 metric, respectively. The improvements demonstrate the generalization ability of EMCL.

**Effect of the scale factor** $\beta$ The parameter $\beta$ is the scale factor. By adjusting $\beta$, we can add some flexibility in the reconstructed video-and-language representations. We evaluate the scale range setting $\beta \in [0.0, 4.0]$ as shown in Table 2. We find that R@1 is improved from 45.1% to 51.6% when $\beta = 2.0$ and saturated with $\beta = 3.0$. As a result, we adopt $\beta = 3.0$ to achieve the best performance.

**Generalize to other contrastive learning tasks** To further verify the generalization of our method, we adopt the EMCL to other contrastive learning tasks, such as Self-supervised Visual representation and Few-shot Classification. Table 3 shows that our EMCL can be applied to boost other contrastive

**Query 7060**: a guy extinguishes something then talks into the camera with another guy.

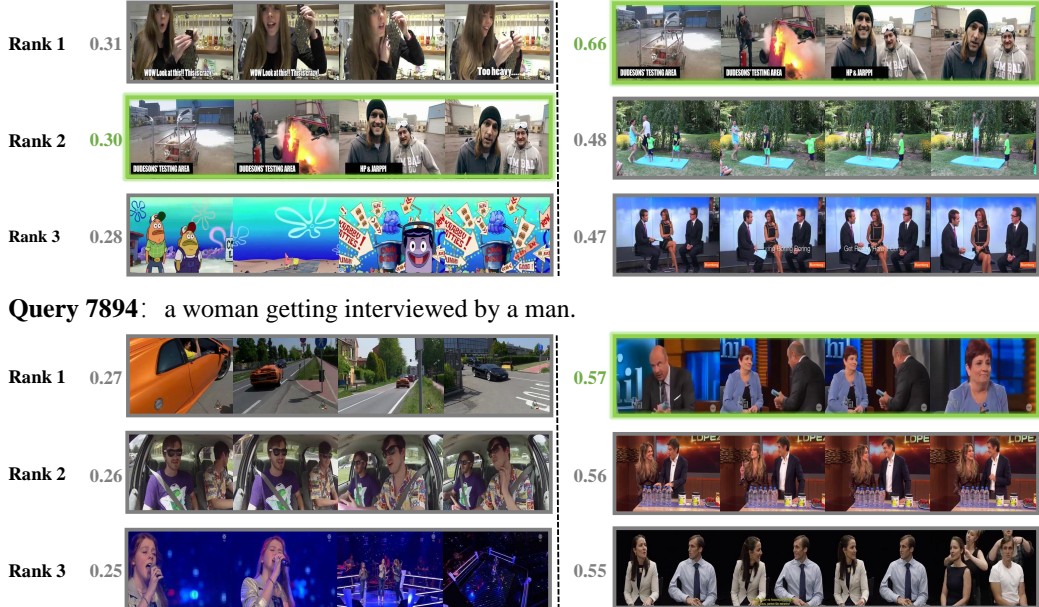

**Query 7894**: a woman getting interviewed by a man.

Figure 2: Visualization of the text-to-video results. The left are the videos ranked by the base model, and the right are the results from our EMCL-Net. Only the correct videos are highlighted in green. We report the similarity between the texts and the videos on the left of the videos.

learning tasks successfully. For the Self-supervised Visual representation, our approach can boost the baseline with an improvement up to 0.16%. For the few-shot classification, our approach with EMCL outperforms the baseline by 0.2%. It shows that EMCL can further improve the performance of contrastive learning, especially for the tasks with similar samples containing redundant dimensions and unable to maintain a large number of negative samples.

**The generalization of our method for long videos**    To further verify the generalization of our method for long videos, we adopt the EMCL to the zero-shot long video classification on the ActivityNet dataset. For the zero-shot long video classification, we use Prompt to transform the video classification task into a video text matching task. The template is "human action of <label>". Table 3 shows that the EMCL can be beneficial to the zero-shot long video classification. The reason may be that for the long video dataset, most of the rich information in the video is redundant information irrelevant to the task. Therefore, by eliminating the redundancy between modalities, we can reduce the interference of noise information and improve the performance of the model.

## D   Additional Qualitative Analysis

**Visualization of semantic similarity in semantic subspace**    In EMCL, we project video and text features into semantic subspaces. We hope that videos and texts with similar semantics can share a common semantic center. In Figure 1, we associate all video frames with words in the text. As shown in Figure 1, all video frames assigned to "skateboard" contain content related to "skateboard". An interesting observation is that in addition to the semantic information, quantitative information such as "person" and "others" are also distinguished in semantic subspace. For example, all video frames assigned to "person" contain only one person, while video frames assigned to "others" contain more than one person. All the semantic similarities in Figure 1 are low because limited training data are not sufficient to understand low-frequency words. This experiment shows that the EMCL can help the model learn adequate semantic information in video-text retrieval.

**Visualization of the text-to-video retrieval**    We show two examples of the videos retrieved by the base model and our EMCL-Net. As shown in Figure 2, our EMCL-Net successfully retrieves the ground-truth video while the model without the EMCL module returns several videos that are unrelated to the query sentence. Our EMCL-Net retrieves the correct videos with higher confidence

than the method without the EMCL module. These results demonstrate that our EMCL module can effectively improve the retrieval performance of the model.