# OpenReview forum: "Expectation-Maximization Contrastive Learning for Compact Video-and-Language Representations"
_NeurIPS.cc/2022/Conference — NeurIPS 2022 Accept_

### Official Review · Reviewer_yWjj · 2022-07-11

**Rating:** 6
**Confidence:** 4
**Soundness:** 3 good
**Presentation:** 3 good
**Contribution:** 2 fair

**Summary:**

The paper presents an approach that improves cross-modal contrastive learning by jointly estimating more compact sub-spaces for both modalities using the well-known EM algorithm. The authors show improvements on three video-text retrieval tasks, and multiple baseline models, to which the proposed sub-space embedding method is being applied. The proposed method works by linearly interpolating the original features with some that have been reconstructed in a shared space, generated using EM estimation with radial Gaussians. This improves similarity between the two spaces, and AFAIK represents a novel idea. For efficient implementation, the authors transfer GMM initializations across mini-batches, which further improves performance.


**Questions:**

I am wondering if and how diagonal or full covariance Gaussians would affect the results? Would the results improve further, or would they deteriorate because the subspace fits better to the data, and therefore less dimensionality reduction and mapping is taking place?
Have you experimented with not just using a single codebook in a video, but multiple codebooks, so that e.g. one could cover indoor and one outdoor, one does cartoons, one does real footage, etc?
Are there any architectures on which the method does not improve? I.e. is there something to learn about which methods does EMCL improve upon, and which ones it does not?


**Limitations:**

The authors work in an established framework, therefore I think there are no specific limitations to discuss.

**Strengths And Weaknesses:**

Strengths

The paper presents a wealth of results and meaningful ablations. Authors include the code in the submission, and commit to making it available (please provide a clear license) upon acceptance of the paper.
In addition to text-video contrastive learning, the authors apply their algorithm to other tasks such as visual representation learning and w shot classification, and observe similar improvement.s

Weaknesses

The paper does not discuss computational requirements, other than stating that the algorithm has constant complexity.
The writing is sometimes a bit awkward, as in "our EMCL", but generally accessible.
The paper does not demonstrate improvements on the latest SOTA (but given the number of different architectures presented, it seems likely it would give improvements)

---

> ### Author Response · Authors · 2022-08-02
> **Response to Reviewer yWjj (1/2)**
>
> We sincerely thank you for your helpful comments! If you have further questions, please feel free to contact us.
>
> > **Q1**: If and how diagonal or full covariance Gaussians would affect the results? Would the results improve further, or would they deteriorate because the subspace fits better to the data, and therefore less dimensionality reduction and mapping is taking place?
>
> **A1**: The diagonal or full covariance Gaussians would reduce the performance.
>
> For **diagonal covariance Gaussians**, the following table shows that diagonal covariance Gaussians will reduce the performance. The reason is that diagonal covariance Gaussians can easily overfit the data.
>
> For **full covariance Gaussians**, we show that full covariance Gaussians and diagonal covariance Gaussians are equivalent in neural networks. Therefore, we only show the experiment result with diagonal covariance Gaussians.
>
> *Detailed Proof: Let us consider a feature $\pmb{Z}$ that follows a joint Gaussian distribution, i.e., $\pmb{Z}\sim\mathcal{N}(\pmb{\mu},\pmb{\Sigma})$. For the real symmetric matrix $\pmb{\Sigma}$, there exists a real orthogonal matrix $\pmb{Q}$, such that $\pmb{\Lambda}=\pmb{Q}^T\pmb{\Sigma}\pmb{Q}$ is a diagonal matrix. Therefore, there exists a linear projection matrix $\pmb{Q}$, such that $\pmb{Q}\pmb{Z}\sim \mathcal{N}(\pmb{Q}\pmb{\mu},\pmb{\Lambda})$. In neural networks, the projection matrix $\pmb{Q}$ is implicitly included in the feature encoder, so full covariance Gaussians and diagonal covariance Gaussians are equivalent.*
>
> |                Methods                 | Text-to-Video R@1 | Video-to-Text R@1 |
> | :------------------------------------: | :---------------: | :---------------: |
> |               Base Model               |       42.4        |       40.4        |
> | + diagonal (full) covariance Gaussians |       46.2        |       47.7        |
> |                 + EMCL                 |     **51.6**      |     **51.8**      |
>
> *Implementation Details: We estimate the variance $\pmb{\sigma}=[\sigma_1, \sigma_2, ..., \sigma_D]$ of each coding bit separately. In step E, the estimated value of $y_{j,k}$ is obtained by $y_{j,k} = \frac{p(x_{:,j},\lambda_{:,k})}{\sum_{k=1}^K p(x_{:,j},\lambda_{:,k})}$. In the M step, we update the variance $\pmb{\sigma}$ by $\hat \sigma_k=\frac{1}{2B}\sum_{i=1}^{2B}\frac{\sum_{j=1}^D y_{j,k}(x_{i,j}-\overline{x_{:,j}})^T(x_{i,j}-\overline{x_{:,j}})}{\sum_{j=1}^D y_{j,k}}$, where $\overline{x_{:,j}}=\frac{1}{2B}\sum_{i=1}^{2B}x_{i,j}$.  Rather than randomly initializing $\pmb{\sigma}$ for each batch, we update $\pmb{\sigma}$ using an average moving method, which is formulated as $\sigma_{k}^{t} = \alpha \sigma_{k}^{t-1}+(1-\alpha) \hat \sigma_{k}^{t}$.*
>
> > **Q2**: Have you experimented with not just using a single codebook in a video, but multiple codebooks, so that e.g. one could cover indoor and one outdoor, one does cartoons, one does real footage, etc?
>
> **A2**: Thank you for bringing this to our attention! We agree that this would be an interesting task. The challenge is the lack of such fine-grained and costly annotations in existing datasets.
>
> > **Q3**: Are there any architectures on which the method does not improve? I.e. is there something to learn about which methods does EMCL improve upon, and which ones it does not?
>
> **A3**: The gain of our method for the single-tower structure is lower than that for the two-tower structure.
>
> The reason is that the joint coding of different modalities can reduce the modality gap to a certain extent in the single tower structure. Therefore, the gain of our method for the single-tower structure is lower than that for the two-tower structure.
>
> On the other hand, although the single-tower structure can alleviate the problem of the modality gap, it cannot construct low-rank representation space and remove the redundancy in the features. Therefore, our method still brings additional improvement over the single-tower structure.
>
> |           Methods           | Model structure | Text-to-Video R@1 | Video-to-Text R@1 |
> | :-------------------------: | :-------------: | :---------------: | :---------------: |
> |    CLIP4clip-tightTransf    |  single tower   |       40.2        |       40.6        |
> | CLIP4clip-tightTransf+ EMCL |  single tower   |    41.0 (+0.8)    |    41.4 (+0.8)    |
> |     CLIP4clip-seqTransf     |   two towers    |       43.4        |       42.4        |
> | CLIP4clip-seqTransf + EMCL  |   two towers    |    46.9 (+3.5)    |    46.6 (+4.2)    |

---

> > ### Author Response · Authors · 2022-08-02
> > **Response to Reviewer yWjj (2/2)**
> >
> > > **Q4**: The paper does not discuss computational requirements, other than stating that the algorithm has constant complexity.
> >
> > **A4**: Our method consumes very little computational resources. As shown in the following table, the EMCL module does not significantly improve the training time, which is mainly for two reasons:
> >
> > * No parameters in the EMCL module need to be updated by the BP (Back Propagation) algorithm.
> > * Since the EMCL module is all composed of matrix operations, we can put it on GPU for acceleration.
> >
> > |     Methods     | epochs | GPU hours (V100) |
> > | :-------------: | :----: | :--------------: |
> > |   Base Model    |   5    |       7.2        |
> > | Base Model+EMCL |   5    |       7.4        |

---

> > > ### Comment · Reviewer_yWjj · 2022-08-09
> > > **Thank you!**
> > >
> > > This makes sense, thank you for running these ablations, and providing explanations.
> > >
> > > Did you optimize the number of parameters in the diagonal vs full covariance case? The model may need fewer diagonal than radial Gaussians (to avoid overfitting), but still the results may further improve. Running this experiment however may become a nit.
> > >
> > > Regarding the multiple codebooks: the model should be able to learn them automatically using simple EM - i.e. you can first compute the expectation for each codebook, add one training example to the best fitting one, and then update its parameters accordingly. Repeat this process next each epoch. This should be able to then support multiple semantic cases, without requiring annotations.

---

> > > > ### Author Response · Authors · 2022-08-10
> > > > **Thanks very much for raising further advice**
> > > >
> > > > Thanks for reading our response and raising further advice.
> > > >
> > > > > **Q1**: Did you optimize the number of parameters in the diagonal vs full covariance case? The model may need fewer diagonal than radial Gaussians (to avoid overfitting), but still the results may further improve.
> > > >
> > > > We didn't optimize the number of parameters. Thank you for bringing this to our attention! We were surprised that the results improved further when we reduced the number of parameters in Gaussians. Thanks again for making our results even stronger!
> > > >
> > > > | The number of parameters | Text-to-Video R@1 | Video-to-Text R@1 |
> > > > | :----------------------: | :---------------: | :---------------: |
> > > > |            32            |       46.2        |       47.7        |
> > > > |            16            |       46.7        |       47.3        |
> > > > |            8             |     **51.4**      |     **51.2**      |
> > > > |            4             |       44.5        |       43.5        |
> > > >
> > > > > **Q2**: Regarding the multiple codebooks: the model should be able to learn them automatically using simple EM.
> > > >
> > > > Thank you for helping us achieve such an interesting idea! According to your comments, we have extended our method to multiple codebooks. Specifically, we add a new dimension *m* to the hidden variables and bases corresponding to *m* codebooks.
> > > >
> > > > We did a straightforward experiment and did not optimize the parameters. We found it more difficult for the model to learn multiple codebooks than a single codebook. In the future, we will try to optimize the method and parameters to solve this problem.
> > > >
> > > > | Methods | Text-to-Video R@1 | Video-to-Text R@1 |
> > > > | :-----------: | :---------------: | :---------------: |
> > > > |  Base Model   |       42.4        |       40.4        |
> > > > | + 2 codebooks |       38.3        |       41.4        |
> > > >
> > > > We sincerely thank you for your helpful comments! We will add the above important discussions in the final manuscript and highlight them. Thanks again for spending a huge amount of time on our paper.

---

### Official Review · Reviewer_widu · 2022-07-11

**Rating:** 6
**Confidence:** 3
**Soundness:** 3 good
**Presentation:** 3 good
**Contribution:** 2 fair

**Summary:**

The paper proposes Expectation-Maximization Contrastive Learning (EMCL) to reduce the modality gap exists in multi-modal representations. The EM operation is applied offline before contrastive learning, making EM have low computation cost and a plug-and-play module. The visualization demonstrates this algorithm's ability to close the modality gap, and the video-text retrieval studies indicate how effective it is.

**Questions:**

1. Is the ECML module trained with the neural network or frozen during contrastive learning? My current understanding is that ECML determines bases offline before the contrastive learning, but lines 69 - 70 say it can also be a jointly training layer, which makes me a bit confused, so it would be great if you could elaborate on this.
2. Follow the previous question, if the module is always frozen, does it mean that the vision and language encoders can not be trained when fine-tuning? If yes, this might limit the usage of ECML since end-to-end training of the whole model is the mainstream approach for now, and it usually leads to better performance.
3. If the EM part can train with contrastive learning, then what are the computational overheads of it (how much extra training time is needed)?
4. I understand that the modality gap seems to be an issue in current multi-modal models. However, I am not sure forcing the data pair to have close representation is always a good thing, especially when the data has complex and mixed information. For example, videos contain rich information, even though some datasets have text labels to describe each video, the text labels may not cover all contents of the video. Then we will lose much information if we limit video representations close to their text label representations, and this might be harmful in some tasks. What is your thought on this?
5. Why EM can group the representation of videos and their labels into the same clusters? From the paper, I figure EM is used as a dimension reduction method, which can remove some redundant information from the data, but I do not get why removing redundancy leads to reducing the modality gap and learning a strong shared representation space. I think they are two things. And also, does learning the compact representation guarantee reducing the modality gap?
6. What are the model sizes of the models and pre-trained weights they use in Table 1? I think the information is necessary for a fair comparison.

**Limitations:**

The paper mentions that the initialization and hyper-parameter choices highly affect EM's performance. I also think the computational overhead (if EM trains with contrastive learning) and the strong assumption of pair data having similar representation might limit the proposed method to extend to other applications.

**Strengths And Weaknesses:**

1. Pros
* The paper introduces an effective EM module that can help the model learn compact representations and reduce the modality gap.
* The EM module can easily incorporate into existing contrastive learning frameworks.
* The experiment session shows strong results, and the ablation studies discuss the design choices of the proposed method completely.

2. Cons
* I am a bit concern about the motivation of "reducing modality gap" (See Questions 4 and 5).
* Some literature might be missing in related works. I think the problem that this paper is interested in (the modality gap) is also highly related to transfer learning and domain adaptation. However, there is no such discussion in the related works.

---

> ### Author Response · Authors · 2022-08-02
> **Response to Reviewer widu (1/2)**
>
> We sincerely thank you for your helpful comments! If you have further questions, please feel free to contact us.
>
> > **Q1**: Is the EMCL module trained with the neural network or frozen during contrastive learning?
>
> **A1**: The EMCL module is trained with the neural network.  The training details are as follows:
>
> * During the training stage, we use a residual structure that preserves the original embeddings along with the new embeddings from our EMCL module. The residual structure enables the vision and language encoders to be trained when fine-tuning. At the same time, we update the initial value $\pmb{M}$ using an average moving method. Each time a new batch of features is fed into the model, we use $\pmb{M}$ to initialize the $\pmb{\lambda}$. Then the components $\pmb{Y}$ and $\pmb{\lambda}$ are updated by iteration.
> * During the inference stage, given a set of queries (text/video) and a set of candidates (videos/texts), we use a trained $\pmb{M}$ to initialize the $\pmb{\lambda}$. Then $\pmb{Y}$ and $\pmb{\lambda}$ can be updated in an unsupervised way by iteration.
> * When the EMCL module is incorporated into trained baselines as an out-of-the-box inference module with no extra training, $\pmb{\lambda}$ is randomly initialized. Then the components $\pmb{Y}$ and $\pmb{\lambda}$ are updated by iteration.
>
> > **Q2**: If the module is always frozen, does it mean that the vision and language encoders can not be trained when fine-tuning?
>
> **A2**: The residual structure enables the vision and language encoders to be trained when fine-tuning.
>
> Specifically, during training, the $\pmb{X}$ in $\hat{\pmb{X}}=\beta f_\text{EMCL}(\pmb{X}) + \pmb{X}$ will back-propagated to the visual and textual encoders. Therefore, the parameters in visual and textual encoders will be fine-tuned during training.
>
> > **Q3**: If the EM part can train with contrastive learning, then what are the computational overheads of it (how much extra training time is needed)?
>
> **A3**: As shown in the following table, the EMCL module does not significantly improve the training time, which is mainly for two reasons:
>
> * No parameters in the EMCL module need to be updated by the BP (Back Propagation) algorithm.
> * Since the EMCL module is all composed of matrix operations, we can put it on GPU for acceleration.
>
> |     Methods     | epochs | GPU hours (V100) |
> | :-------------: | :----: | :--------------: |
> |   Base Model    |   5    |       7.2        |
> | Base Model+EMCL |   5    |       7.4        |
>
> > **Q4**: We will lose much information if we limit video representations close to their text label representations, and this might be harmful in some tasks. What is your thought on this?
>
> **A4**: We agree that there is indeed a tradeoff between bridging the gap between modalities and maintaining the diversity of representation. Our method also takes this into account.
>
> Specifically, we use the residual structure to retain this discriminative information and use scale factor $\beta$ to balance the original and new embeddings. By adjusting $\beta$, we can add flexibility to the reconstructed video-and-language representations for various downstream tasks. In Table 6, we show the experiments on the influence of the parameter $\beta$.
>
> However, reducing the modality gap is a meaningful exploration direction if we can eliminate the noise of text labels. Besides, recent work [1] points out that reducing the modality gap helps improve zero-shot performances and reduce the racial bias of neural networks. We continue to believe that reducing the modality gap is an interesting and worthwhile direction to explore.
>
> [1] Liang, Weixin, et al. "Mind the gap: Understanding the modality gap in multi-modal contrastive representation learning." arXiv preprint arXiv:2203.02053 (2022).

---

> > ### Author Response · Authors · 2022-08-02
> > **Response to Reviewer widu (2/2)**
> >
> > > **Q5**: Why removing redundancy leads to reducing the modality gap and learning a strong shared representation space?  Does learning the compact representation guarantee reducing the modality gap?
> >
> > **A5**: The main reason is that we use the EM algorithm to remove the redundancy of the **concatenation** $\pmb{X}= [\pmb{C_v};\pmb{C_t}]$ of visual embedding and textual embedding, thus reducing the modality gap. The explanation is as follows:
> >
> > * Firstly, the EMCL module accepts $\pmb{X}= [\pmb{C_v};\pmb{C_t}]$ as input which is a concatenation of visual embedding and textual embedding.  Our method decomposes $\pmb{X}$ to obtain the **shared** variable $\pmb{Y}\in\mathbb{R}^{D\times K}$ of **video features** and **text features**. It forces the two modalities to share the common embedding space, where both video and text features are represented with the distributions over the same set of hidden variables. Then contrastive learning on these representations can bridge the modality gap and preserve strong semantic alignment across modalities.
> > * Secondly, learning the compact representation forces video and text to be represented as lower rank representations in compact and semantically related subspaces. The low-rank property can help reduce redundancy between modalities and intra-class variance and therefore help reduce the modality gap. Moreover, the compact representation promotes the discriminating ability of the reconstructed representation.
> >
> > Therefore, our method explicitly forces both modalities to be represented in the same space. At the same time, our method leverages these compact features for contrastive learning to improve their performance efficiently. Fig. 1 and Fig. 4-5 can prove it.
> >
> > > **Q6**: What are the model sizes of the models and pre-trained weights they use in Table 1? I think the information is necessary for a fair comparison.
> >
> > **A6**: Thanks for this suggestion! We have added the pre-trained weights of the models in Table 1. The pre-training weights and the number of pre-training parameters used by each model are summarized as follows:
> >
> > |   Methods   | Pre-trained weights | Pre-training parameters |
> > | :---------: | :-----------------: | :---------------------: |
> > |     CE      |        GPT-1        |          117M           |
> > |     MMT     |      BERT-Base      |           84M           |
> > | Support-Set |       T5-Base       |          220M           |
> > |   T2VLAD    |      BERT-Base      |           84M           |
> > |  CLIP4Clip  |   CLIP (ViT-B/32)   |          151M           |
> > |    Ours     |   CLIP (ViT-B/32)   |          151M           |
> >
> > > **Q7**: The problem that this paper is interested in (the modality gap) is also highly related to transfer learning and domain adaptation.
> >
> > **A7**: We have added the discussion and several related works on transfer learning [1,2] and domain adaptation [3,4] in the section "Related Works".
> >
> > [1] Phung, Trung, et al. "On Learning Domain-Invariant Representations for Transfer Learning with Multiple Sources." NeurIPS (2021): 27720-27733.
> >
> > [2] Neyshabur, Behnam, Hanie Sedghi, and Chiyuan Zhang. "What is being transferred in transfer learning?." NeurIPS (2020): 512-523.
> >
> > [3] Liang, Jian, et al. "Pareto domain adaptation." NeurIPS (2021): 12917-12929.
> >
> > [4] Stojanov, Petar, et al. "Domain adaptation with invariant representation learning: What transformations to learn?." NeurIPS (2021): 24791-24803.

---

> > > ### Comment · Reviewer_widu · 2022-08-07
> > > **Additional thoughts on the new responses**
> > >
> > > Thanks for your replies.
> > >
> > > I only have one additional thought on **A4**.
> > >
> > > > However, reducing the modality gap is a meaningful exploration direction if we can eliminate the noise of text labels.
> > >
> > > I agree. However, I am afraid this might never happen especially since the targeted tasks are about video and text. Both sources have rich information, and it is likely one source can only describe part of the other source because it is not realistic to require annotators to annotate EVERY detail of the video. That is why there might be some downsides to applying ECML to some tasks in my opinion. The incomplete annotations might happen more commonly for long-form videos, and reducing the modality gap might have different influences on different tasks, so I think it would be helpful for the authors to apply ECML to long video datasets and broader tasks in the future.

---

> > > > ### Author Response · Authors · 2022-08-08
> > > > **Thanks very much for raising further advice**
> > > >
> > > > Thanks very much for reading our response and raising further advice!
> > > >
> > > > Note that *ActivityNet* used in our experiment is a popular large-scale long video dataset [1]. It consists of 20K YouTube videos, where the average video length is 180s, and the average text length is 54 words.
> > > > As shown in Table 1 and Table 5, EMCL can be applied to the long video dataset (i.e., the *ActivityNet Captions* dataset) for the video retrieval task.
> > > >
> > > > To further verify the generalization of our method, we adopt the EMCL to zero-shot long video classification on the *ActivityNet* dataset. The following table shows that the EMCL can also be applied to the zero-shot long video classification.
> > > >
> > > > |    Methods     |    Top1 Acc     |    Top5 Acc     |
> > > > | :------------: | :-------------: | :-------------: |
> > > > |   Clip4clip    |      51.7       |      78.5       |
> > > > | Clip4clip+EMCL | **52.3 (+0.6)** | **79.0 (+0.5)** |
> > > >
> > > > *Implementation Details:  We use Prompt to transform the video classification task into a video text matching task. The template is "human action of <label>".*
> > > >
> > > > Therefore, reducing the modality gap is still beneficial for long video datasets. The reason may be that for the long video dataset, most of the rich information in the video is redundant information irrelevant to the task. Therefore, by eliminating the redundancy between modalities, we can reduce the interference of noise information and improve the performance of the model.
> > > >
> > > > We sincerely thank you for your helpful comments!  We will add this important discussion in the final manuscript and highlight it. We hope to have addressed your concern, and thanks again for spending a huge amount of time on our paper.
> > > >
> > > > [1] EclipSE: Efficient Long-range Video Retrieval using Sight and Sound, ECCV 2022.

---

> > > > > ### Comment · Reviewer_widu · 2022-08-09
> > > > > **Final thoughts and scores**
> > > > >
> > > > > Thanks for the additional experiments and for addressing my questions.
> > > > >
> > > > > It would be interesting to see if ECML can be applied to broader tasks, such as QA, captioning, and reasoning in the future. This paper will have a bigger contribution if having those results.
> > > > >
> > > > > The updated draft helps the presentation of the paper, so I will increase the presentation score to 3. Also, given the efficiency and effectiveness of the proposed method, I will increase the overall to 6.
> > > > >
> > > > > Thanks again for your time during the discussion.

---

> > > > > > ### Author Response · Authors · 2022-08-10
> > > > > > **Thank you!**
> > > > > >
> > > > > > Thanks again for your valuable suggestions and comments. We really enjoy communicating with you and appreciate your efforts.

---

### Official Review · Reviewer_f2Q1 · 2022-07-11

**Rating:** 6
**Confidence:** 4
**Soundness:** 3 good
**Presentation:** 3 good
**Contribution:** 3 good

**Summary:**

This paper addresses the task of bidirectional text-to-video retrieval and the limitation of contrastive learning that is commonly used to learn discriminative visual and text features in a shared latent space. The authors qualitatively show that state-of-the-art approaches still do not learn a strong alignment between the visual and text modalities and propose a reformulation of the standard contrastive learning as an expectation-maximization project to bridge the gap between both modalities. Finally, the paper introduces a parameter initialization strategy for the proposed expectation-maximization approach, which results in significant improvements on the task across three commonly-used datasets.

**Questions:**

It is not very clear from the writing why learning low rank representations will lead to a stronger alignment between both modalities in the shared embedding space. This might be out of scope, but a proof may be helpful in conveying this intuition.

**Limitations:**

1) Line 128: y_{j,k} should be y_{i,j,k} instead? The i-th index is not used in this notation.

**Strengths And Weaknesses:**

Strengths:

1) The paper is largely well-written and easy to understand. In particular, the mathematical definitions that are provided are very helpful for understanding the proposed approach.

2) A known limitation of contrastive learning is that it requires a large number of negative samples during training to learn discriminative features that are effective for bidirectional text-to-video retrieval. Furthermore, these negative samples have to be hard negatives, which can be very difficult to discover without fine-grained and costly annotations. The proposed expectation-maximization approach helps to mitigate this issue to a large degree by learning a low-rank shared latent feature space, which removes redundancy. Interestingly, this results in a much stronger alignment between the learnt visual and text representations. As such, this is a significant contribution to this task and multimodal representation in general. While the expectation-maximization algorithm is not new, its application to the problem of learning more compact and effective multimodal representations is original.

3) The authors also conducted extensive ablation experiments over the variables including the number of subspaces as well as number of iterations used in the expectation-maximization algorithm. These are especially helpful for the reader to quickly determine the contributions of each component of the proposed approach.

Weaknesses:

1) Some of the figures are pretty small and the numbers can be a little hard to read, even in the magnified state.

---

> ### Author Response · Authors · 2022-08-02
> **Response to Reviewer f2Q1**
>
> We sincerely thank you for your helpful comments! If you have further questions, please feel free to contact us.
>
> > **Q1**: It is not very clear from the writing why learning low rank representations will lead to a stronger alignment between both modalities in the shared embedding space.
>
> **A1**: The main reason is that we use the EM algorithm to learn the low-rank representation of the **concatenation** $\pmb{X}= [\pmb{C_v};\pmb{C_t}]$ of visual embedding and textual embedding, thus reducing the modality gap. The explanation is as follows:
>
> * Firstly, the EMCL module accepts $\pmb{X}= [\pmb{C_v};\pmb{C_t}]$ as input which is **a concatenation of visual embedding and textual embedding**.  Our method decomposes $\pmb{X}$ to obtain the **shared** variable $\pmb{Y}\in\mathbb{R}^{D\times K}$  of **video features** and **text features**. It forces the two modalities to share the common embedding space, where both video and text features are represented with the distributions over the same set of hidden variables. Then contrastive learning on these representations can bridge the modality gap and preserve strong semantic alignment across modalities.
> * Secondly, learning the compact representation forces video and text to be represented as lower rank representations in compact and semantically related subspaces. The low-rank property can help reduce intra-class variance and therefore help reduce the modality gap. Moreover, the compact representation promotes the discriminating ability of the reconstructed representation.
>
> Therefore, our method explicitly forces both modalities to be represented in the same space. At the same time, our method helps contrastive learning to learn discriminative features. Fig. 1 and Fig. 4-5 can prove it.
>
> > **Q2**: Line 128: $y_{j,k}$ should be $y_{i,j,k}$ instead? The $i_{th}$ index is not used in this notation.
>
> **A2**: The introduced parameter $\pmb{Y}$ for EMCL is in the size of $D\times K$, it projects the dimension of visual and textual concatenation $2B\times D$ into the dimension of the semantic space $2B \times K$. This parameter $\pmb{Y}$ is irrelevant to the instances in each batch, therefore, there is no need to introduce the index $i$ into the variable $y_{j,k}$.
>
> > **Q3**: Some of the figures are pretty small and the numbers can be a little hard to read, even in the magnified state.
>
> **A3**: Thanks for this suggestion! We have scaled up all the figures and the fonts in the figures in the revision.

---

### Official Review · Reviewer_jsU2 · 2022-07-13

**Rating:** 4
**Confidence:** 3
**Soundness:** 2 fair
**Presentation:** 2 fair
**Contribution:** 2 fair

**Summary:**

Video-and-Language representation learning methods represent the data from both modalities into a shared latent space. However due to separate independent encoders for the two modalities, there is a gap between the representations of the two modalities. To bridge the modality gap, this paper proposes to learn a shared latent space by representing the data from both the modalities into the same latent space defined by basis vectors that are learned using an EM style algorithm. The paper evaluates the proposed approach and the baselines on three text-video retrieval benchmark datasets.

**Questions:**

Questions:

1. As pointed out in Weakness 1, the modality gap phenomenon is arising because the representation heads/encoders are separate for the modality in the CLIP4Clip approach. This results in two independent spaces and the contrastive loss tries to reduce the distance for a given pair of text and video. Even though the contrastive loss reduces the distances for many such pairs of text and video, it still lacks in reducing the distance between the two latent spaces. However, passing the representations from the two encoders through a common set of nonlinear layers can explicitly project the data from two modalities into a shared space. Applying the contrastive loss on top of it can make the embeddings of the text-image pair to be closer to each other in this shared space while explicitly forcing the two modalities to share the common latent space. This simplistic approach is analogous to the proposed approach and thus should be compared as a baseline.
2. Lines 186-191, the paper describes the plug-n-play module which misses a key detail. Without training EMCL, how are the components Y and \lambda learned? I think the EMCL model needs to be trained to get the values of these variables. The results in the section “Generalization Analysis” also don’t shed any light into the method of plugging in the EMCL module.
3. Continuing on Question 1, in lines 186-191 the alternatively be implemented using a Residual layer that preserves the original embeddings along with the new embeddings from the shared space.
In the section “Comparisons to other Dimensionality Reduction Methods”, the experiment setup is not clearly explained. What is “Linear”?

**Strengths And Weaknesses:**

Strengths:
1. The paper points out a very important phenomenon of the modality gap when representing the data into a shared latent space.
2. The paper proposes a way to represent the modalities into a shared latent space of learned basis vectors.
3. The proposed method is a plug n play framework that can be applied to any representation learning algorithm
4. The results show an improvement in performance when the proposed method is added to standard representation learning methods.

Weakness:
1. The paper adopts a very complicated approach to address the phenomenon of modality gap, by selecting a set of basis vectors and then representing the data using a linear combination of those vectors. A simplistic baseline can be passing the representations through a common feed-forward network which explicitly forces the modality to be represented in the same space. The comparison to such a simplistic baseline is missing.
2. The proposed approach has a few critical hyperparameters (number of basis vectors, number of EM steps) which if not properly set can degrade the performance.
3. The writing of the method description lacks clarity with some aspects of the proposed approach not explained clearly.
The intuition behind selecting the baselines are not explained. Moreover, the baselines are not explained at all. Mere mentioning the names of the baselines in the Table significantly reduces the readability of the paper.

In summary, the paper points out to an important phenomenon, although the proposed method with extra hyperparameters need to be compared to relevant simple baselines. In addition, the writing needs to be greatly improved by including proper explanation of baselines, the intuition behind choosing them and clear explanations for all the aspects of the proposed method.

---

> ### Author Response · Authors · 2022-08-02
> **Response to Reviewer jsU2 (1/2)**
>
> We sincerely thank you for your helpful comments! If you have further questions, please feel free to contact us.
>
> > **Q1**: The proposed method need to be compared to relevant simple baselines, e.g., a shared feed-forward network.
>
> **A1**: Thanks for this suggestion! We compared our method with a shared fully connected layer ("Linear") in the section "Comparisons to other Dimensionality Reduction Methods". As shown in Table 3, our method outperforms this simple baseline.
>
> In addition, we have included two more baselines: (1) Feed-Forward Network (FFN); (2) Transformer. As displayed in the table below, the proposed method outperforms these two baselines.
>
> |        Methods        | Activation Function | Text-to-Video R@1 | Video-to-Text R@1 |
> | :-------------------: | :-----------------: | :---------------: | :---------------: |
> |      Base Model       |       ------        |       42.4        |       43.5        |
> |     + 1 layer FFN     |        Relu         |       42.4        |       42.9        |
> |     + 2 layer FFN     |        Relu         |       43.2        |       43.3        |
> |     + 3 layer FFN     |        Relu         |       44.0        |       43.7        |
> |     + 4 layer FFN     |        Relu         |       44.0        |       42.4        |
> | + 1 layer Transformer |        Gelu         |       41.3        |       40.4        |
> | + 2 layer Transformer |        Gelu         |       41.2        |       40.1        |
> | + 3 layer Transformer |        Gelu         |       40.1        |       39.6        |
> | + 4 layer Transformer |        Gelu         |       40.8        |       41.6        |
> |     + EMCL (ours)     |       ------        |     **51.6**      |     **51.8**      |
>
> Our proposed method has two unique processing steps compared to the common neural network:
>
> * $\pmb{\lambda} \in\mathbb{R}^{2B\times K}$ is a set of bases with lower dimensions, which forces video and text to be represented as lower rank representations. The low-rank property can help reduce intra-class variance and promote the discriminating ability of the reconstructed representation.
> * In this low-rank space, $\pmb{Y}\in\mathbb{R}^{D\times K}$ is the shared variable of **video features** and **text features**, which forces the two modalities to share the common latent space, thus helping contrastive learning to bridge the gap between modalities.
>
> > **Q2**: Lines 186-191, the paper describes the plug-n-play module which misses a key detail. Without training EMCL, how are the components $Y$ and $\lambda$ learned?
>
> **A2**: The parameters $\pmb{Y}$ and $\pmb{\lambda}$ of the EM algorithm can be updated in unsupervised way by iteration without supervised labels, as shown in Algorithm 1.
>
> Specifically, as shown in lines 186-192, the EMCL module accepts $\pmb{X}$ as input which is a concatenation of visual embedding and textual embedding. Then the parameters $\pmb{Y}$ and $\pmb{\lambda}$ of the EM algorithm are updated in an iterative manner according to Algorithm 1.  In step E,  $y_{j,k}$ is updated by $y_{j,k} = \frac{p(x_{:,j},\lambda_{:,k})}{\sum_{k=1}^K p(x_{:,j},\lambda_{:,k})}$, where $p(x_{:,j},\lambda_{:,k}) = \exp{\left(\frac{\sum_{i=1}^{2B} x_{i,j}\lambda_{i,k}}{\sigma}\right)}$. In step M,  $\lambda_{i,k}$ is updated by $\lambda_{i,k} = \frac{\sum_{j=1}^D y_{j,k}x_{i,j}}{\sum_{j=1}^D y_{j,k}}$.
>
> > **Q3**: The results in the section "Generalization Analysis" also don’t shed any light into the method of plugging in the EMCL module.
>
> **A3**: We place the EMCL module after the video and text encoders and retain a residual structure to balance the original and new embeddings.
>
> Since the input and output dimensions of the EMCL module are the same, it is model-agnostic and can be applied to features extracted from any language and video encoders to boost their performances (Table 2).
>
> Specifically, as shown in lines 186-192, the EMCL module accepts $\pmb{X}$ as input which is a concatenation of visual embedding and textual embedding. Then the parameters  $\pmb{Y}$ and $\pmb{\lambda}$ of the EM algorithm are updated in an iterative manner according to Algorithm 1. After several rounds, the refined $\pmb{Y}$ and $\pmb{\lambda}$ are used to reconstruct $\pmb{X}$ into $f_\text{EMCL}(\pmb{X})$. The reconstructed feature and the original feature are weighted together into $\hat{\pmb{X}}=\beta f_\text{EMCL}(\pmb{X}) + \pmb{X}$. Finally, $\hat{\pmb{X}}$ is sent to the following modules.

---

> > ### Author Response · Authors · 2022-08-02
> > **Response to Reviewer jsU2 (2/2)**
> >
> > > **Q4**: In lines 186-191, the alternatively be implemented using a Residual layer that preserves the original embeddings along with the new embeddings from the shared space.
> >
> > **A4**: The residual structure is used to improve the robustness of the model [1].
> >
> > For the cross-modal retrieval tasks, in order to avoid noise, we introduce the EMCL module to reconstruct unified visual and textual features with parametric models which is represented as $f_\text{EMCL}(\pmb{X})$. To retain original semantic information, we use scale factor $\beta$ to balance the original and new embeddings. By adjusting $\beta$, we can add flexibility to the reconstructed video-and-language representations for various downstream tasks.
> >
> > [1] Chun, Sanghyuk, et al. “Probabilistic embeddings for cross-modal retrieval.” CVPR. 2021.
> >
> > > **Q5**: In the section "Comparisons to other Dimensionality Reduction Methods", the proper explanation of baselines and the intuition behind choosing them are not clearly explained. What is "Linear"?
> >
> > **A5**: We chose three baselines, e.g., PCA, "Linear" and "Sparse Autoencoders". The intuition behind choosing them is as follows:
> >
> > * PCA is a popular method for finding salient features. Suppose we input video and text representations simultaneously. In that case, PCA can find the most salient features shared by the two modalities so that the modality gap can be reduced to a certain extent.
> > * "Linear" represents a common set of fully connected layers between two modalities. A common neural network can explicitly project the data from two modalities into a shared space. For clarity, we have changed "Linear" to "Fully connected layers" in Table 3.
> > * "Sparse Autoencoders" represents a common sparse autoencoder between two modalities. Sparse autoencoder reduces the average response of the encoding layer for sparsity, which can find the sparse information shared by the two modalities so that the modality gap can be reduced.
> >
> > According to the experimental results in Table 3, the proposed EMCL outperforms these three baselines.
> >
> > > **Q6**: In the section "Comparisons to other Dimensionality Reduction Methods", the experiment setup is not clearly explained.
> >
> > **A6**: The experiment setup is as follows:
> >
> > For all methods, we concatenate video features $\pmb{C_v}$ and text features $\pmb{C_t}$, generating the input data $\pmb{X} = [\pmb{C_v};\pmb{C_t}] \in\mathbb{R}^{2B\times D}$. Finally, we add reconstructed features $f(\pmb{X})$ and original features $\pmb{X}$ to obtain final text-video representations $\hat{\pmb{X}} = [\hat{\pmb{C_v}};\hat{\pmb{C_t}}] = f(\pmb{X}) + \pmb{X} \in\mathbb{R}^{2B\times D}$. All networks are optimized with the batch size of 128 in 5 epochs.
> >
> > * In "PCA", we adopt PCA at the end of the video-text encoders. We use PCA to reduce the dimensions of the original features from 512 to 32, then restore them to 512 dimensions.
> > * In "Linear", we pass the original features through a common feed-forward network where the inner-layer has a dimensionality of 256, and the activation function is Relu.
> > * In "Sparse Autoencoders", we adopt Sparse Autoencoders at the end of the video-text encoders. The inner-layer has a dimensionality of 256, and the activation function is Sigmoid. We reduce the average response of the encoding layer to $\rho=0.05$.
> >
> > > **Q7**: The writing needs to be improved by including proper explanation of baselines, the intuition behind choosing them and clear explanations for all the aspects of the proposed method.
> >
> > **A7**: Thanks for this suggestion! We have added detailed discussions in the revision. The major updates are as follows:
> >
> > * We have added detailed explanation of baselines in the section "Comparisons to other Dimensionality Reduction
> >   Methods" (lines 232-237) and the detailed experimental setup of baselines in the Appendix (lines 520-528 and 539-550).
> > * We have added more details and explanations for EMCL training and model inference in the section "EMCL-Net" (lines 190-192) and the section "Generalization Analysis" (lines 223-225 and 227).
> > * We have added the details of parameter update rule in the section "Training Objective" (lines 193-196 and 203-205).

---

### Meta-Review · Area_Chair_nJ1n · 2022-08-29

**Recommendation:** Accept
**Confidence:** Certain

**Metareview:**

This paper proposes Expectation-Maximization Contrastive Learning (EMCL) to learn compact video-and-language representations for the general goal of projecting the video and text features into a common latent space. Reviewers agreed that this is an important problem of modality gap, a useful way to represent the modalities into a shared latent space of learned basis vectors, and good improvements. Some reviewers also expressed concerns that the approach is a bit complicated and simpler baselines should be compared to (done in rebuttal); several missing related works discussion w.r.t. distillation, dimensionality reduction, transfer learning/domain adaptation; some motivation confusions about closing representation gap; more tasks should be added such as QA and captioning.


**Award:**

No

---

### Decision · Program_Chairs · 2022-09-14

Accept